# Quantitative perturbation-based analysis of gene expression predicts enhancer activity in early *Drosophila* embryo

Rupinder Sayal[1,2†], Jacqueline M Dresch[3,4†], Irina Pushel[1,5], Benjamin R Taylor[6,7], David N Arnosti[1]*

[1]Department of Biochemistry and Molecular Biology, Michigan State University, East Lansing, United States; [2]Department of Biochemistry, DAV University, Jalandhar, India; [3]Department of Mathematics, Michigan State University, East Lansing, United States; [4]Department of Mathematics and Computer Science, Clark University, Worcester, United States; [5]Stowers Institute for Medical Research, Kansas City, United States; [6]Department of Computer Science and Engineering, Michigan State University, East Lansing, United States; [7]School of Computer Science, Georgia Institute of Technology, Atlanta, United States

**Abstract** Enhancers constitute one of the major components of regulatory machinery of metazoans. Although several genome-wide studies have focused on finding and locating enhancers in the genomes, the fundamental principles governing their internal architecture and *cis*-regulatory grammar remain elusive. Here, we describe an extensive, quantitative perturbation analysis targeting the dorsal-ventral patterning gene regulatory network (GRN) controlled by *Drosophila* NF-κB homolog Dorsal. To understand transcription factor interactions on enhancers, we employed an ensemble of mathematical models, testing effects of cooperativity, repression, and factor potency. Models trained on the dataset correctly predict activity of evolutionarily divergent regulatory regions, providing insights into spatial relationships between repressor and activator binding sites. Importantly, the collective predictions of sets of models were effective at novel enhancer identification and characterization. Our study demonstrates how experimental dataset and modeling can be effectively combined to provide quantitative insights into *cis*-regulatory information on a genome-wide scale.

*For correspondence: arnosti@msu.edu

†These authors contributed equally to this work

**Competing interests:** The authors declare that no competing interests exist.

## Introduction

Developmentally expressed genes in metazoans are regulated by diverse *cis*-regulatory elements, including distally-acting sequences termed enhancers (*Levine, 2010*; *Smith and Shilatifard, 2014*; *Heinz et al., 2015*). Despite more than three decades of progress, surprisingly little is known about constraints on internal structural organization of binding sites ('grammar') within these elements (*Dickel et al., 2013*). Some enhancers show little evolutionary variation, and permit no change in transcription factor binding sites without catastrophic effects on function (*Thanos and Maniatis, 1995*; *Kim and Maniatis, 1997*). Many developmental enhancers, however, demonstrate a more flexible deployment of binding sites, thus functionally conserved elements can exhibit a large degree of evolutionary variation (*Junion et al., 2012*). Although high-throughput studies have dramatically increased our knowledge of genome-wide transcription factor occupancy and transcript expression, we have a limited ability to interpret the functional relevance of quantitative aspects of protein binding or DNA sequence variation. A quantitative understanding of the internal enhancer grammar of *cis*-regulatory elements will provide researchers with powerful tools to better understand the

**eLife digest** DNA contains regions known as genes, which may be "transcribed" to produce the RNA molecules that act as templates for building proteins and regulate cell activity. Proteins called transcription factors can bind to specific sequences of DNA to influence whether nearby genes are transcribed. For example, so-called enhancer regions of DNA contain several binding sites for transcription factors, and this binding activates gene transcription. Little is known about how the transcription factor binding sites are organized in enhancer regions, which makes it difficult to use DNA sequence information alone to predict the regulation of genes.

A transcription factor called Dorsal controls the activity of a network of genes that plays a crucial role in the development of fruit fly embryos. Dorsal binds to the enhancer region of a gene called *rhomboid*, which has been well studied and is known to be a fairly typical example of an enhancer region.

To understand the regulatory information encoded in the DNA sequences of enhancers, Sayal, Dresch et al. have now used a technique called perturbation analysis to investigate the interactions that are likely to occur between Dorsal and other transcription factors as they bind to the *rhomboid* enhancer. This technique involves systematically mutating the enhancer to remove different combinations of transcription factor binding sites and quantitatively investigating the effect this has on gene activity. A large set of mathematical models were then trained using this data and shown to correctly predict the activity of a range of other gene regulatory regions. The collective predictions of the models identified new enhancer regions and revealed details about how different types of transcription factor binding sites are arranged within enhancers.

As we enter an era where the DNA sequences of entire human populations are increasingly accessible, we would like to know the functional significance of changes in gene regulatory regions. Sayal, Dresch et al. show that the regulatory properties of specific control proteins are accessible by employing quantitative experiments and mathematical models. Similar studies will be required to learn how mutations found across the genome may alter gene expression, leading to better diagnosis and treatment of disease.

significance of genetic variation that is observed within and between species, critical for exploitation of burgeoning genomic resources. Thermodynamic models employ tools from statistical physics to model gene activity, providing a framework for understanding transcription factor interactions with specific DNA sequences to regulate gene expression (*Bintu et al., 2005*; *Ay and Arnosti, 2011*; *Dresch and Drewell, 2012*) Previous efforts at thermodynamic modeling in eukaryotic systems have demonstrated that diverse types of data can be fit, providing at least a qualitative level of prediction (*Zinzen et al., 2006*; *Segal et al., 2008*; *He et al., 2010*; *Drewell et al., 2014*). However, earlier studies relied on heterogeneous and low-resolution datasets, inherently limiting modeling effectiveness. In addition, few types of models were tested, reducing the chance that essential properties of transcription factor interactions will be captured.

Here, we tested the hypothesis that an in-depth and quantitative analysis of key transcriptional regulators on an archetypal enhancer would reveal common transcriptional behaviors of these proteins for genome-wide analysis. To harness the potential of thermodynamic modeling approaches, we developed an in-depth enhancer perturbation analysis that takes advantage of the quantitative setting of the *Drosophila* blastoderm embryo. The *rhomboid (rho)* enhancer directs gene expression in the presumptive neuroectoderm under the control of the activator Dorsal, a homolog of NF-κB. The Twist activator and Snail repressor provide additional essential inputs (*Figure 1A*). To extract fundamental information from this data set, we created and fit a comprehensive set of thermodynamic models designed to capture likely interactions between transcription factors as they interact with the enhancer. An extensive set of other coordinately-regulated Dorsal/Twist/Snail target genes was then used to assess the power of this modeling approach for interpretation of *cis*-regulatory variation.

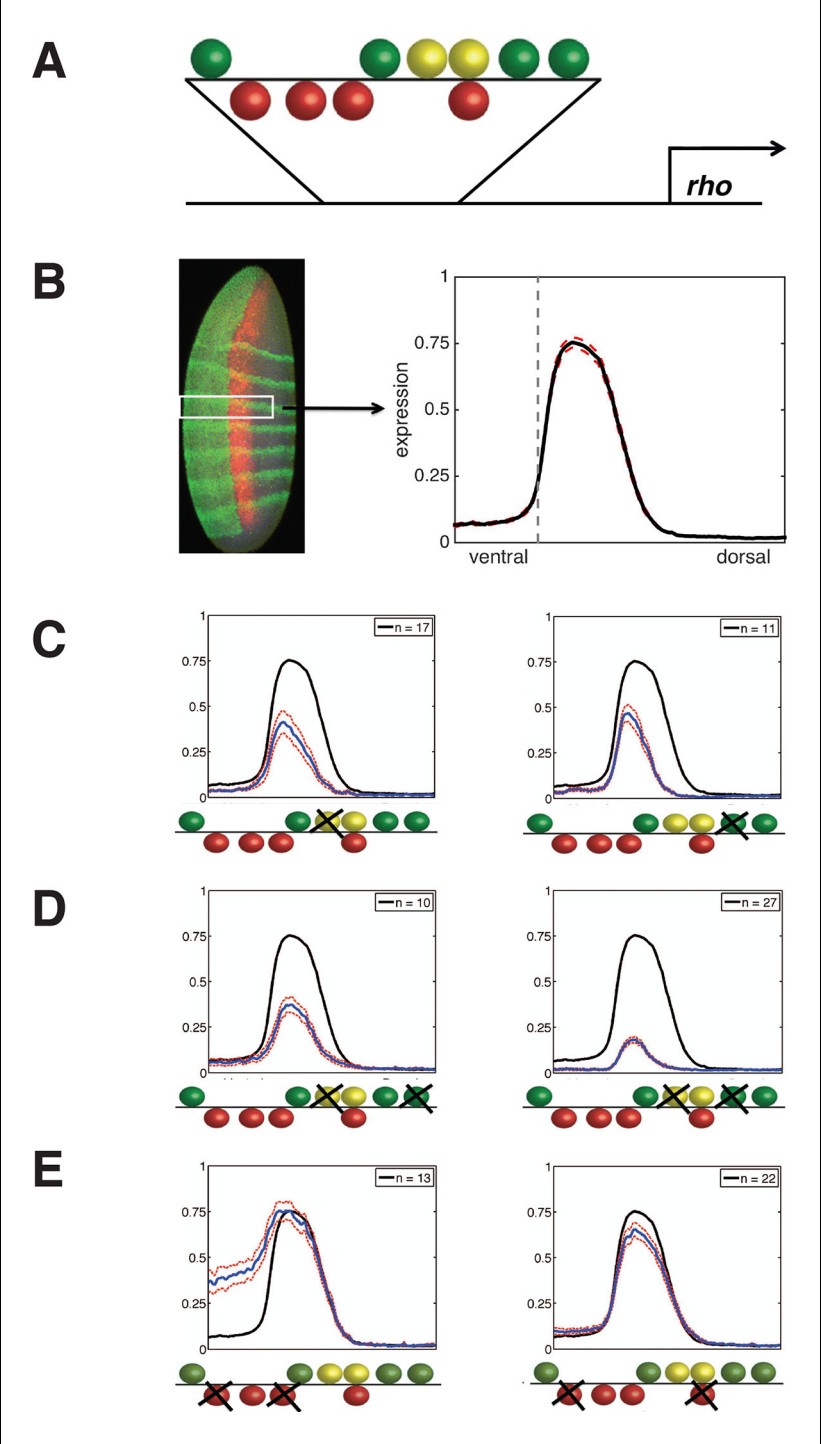

**Figure 1.** Experimental deep perturbation analysis of a neuroectoderm-specific enhancer. (**A**) *rhomboid (rho)* enhancer with footprinted Dorsal activator sites shown in green, Twist activator sites in yellow, and Snail repressor sites in red. (**B**) Drosophila embryo stained to measure *rho* reporter readout in the neuroectoderm with quantitative expression measured using confocal laser scanning microscopy. The embryo is oriented anterior up and dorsal to the right. *rho-lacZ* mRNA is shown in red, seven *even-skipped* mRNA stripes in green, and *snail* mRNA in the mesoderm (left) also in green. Average expression after normalization over embryos containing the wild-type enhancer in the region of interest (white box) is plotted in black at right; dashed red line indicates standard error; vertical dotted line indicating dorsal-most boundary of *sna* expression. (**C**) Effects on expression of removal of single activator sites – expression levels decrease by a similar extent when either Dorsal or Twist

*Figure 1 continued on next page*

*Figure 1 continued*
activator sites are removed. (**D**) Effects of double activator site mutation – expression levels are variably impacted depending on which pairs of sites are affected. (**E**) Effects on expression of removal of pairs of Snail repressor sites, leading to partial depression in ventral regions, or no derepression. For panels **C–E**, solid black lines indicate wild-type enhancer expression; solid blue lines indicate expression of the tested *rho* enhancer; dashed red lines indicate standard error; n = number of embryos imaged for the given construct shown (in top right of each panel).
The following figure supplement is available for figure 1:

**Figure supplement 1.** Quantitative perturbation analysis of *rho* neuroectodermal enhancer.

## Results

### Generation of a quantitative deep perturbation dataset

*rho* is first expressed in two lateral stripes in the presumptive neurogenic ectoderm of the *Drosophila* embryo under cooperative activation by Dorsal and Twist (*Ip et al., 1992*; *Hong et al., 2008*). Expression is excluded from the mesoderm (ventral region) by Snail, a short-range repressor that interferes with activators located within ~100 bp of a Snail binding site (*Gray et al., 1994*). We systematically mutated all activator and repressor binding sites in *rho* neurectodermal enhancer, removing Dorsal or Twist sites individually or in pairs to diminish activation, and removing multiple Snail sites to reduce repression (*Figure 1*, *Figure 1—figure supplement 1*, and *Supplementary file 1*). All 38 enhancers were cloned and integrated into the fly genome using a site-specific integration vector (*Bischof et al., 2007*). We measured the transcriptional output using fluorescent in situ hybridization (FISH) and confocal laser scanning microscopy, and analyzed gene expression data using an image-processing pipeline (*Ay et al., 2008*). Expression data from a total of 935 images - a minimum of ten embryos per construct - was normalized and combined to provide average expression patterns for each enhancer variant (*Figure 1B*, *Figure 1—figure supplement 1*, and *Supplementary file 2*).

Mutation of any single Dorsal or Twist activator binding site resulted in a measurable reduction of peak intensity and retraction of the *rho* stripe from the dorsal region, where activators Dorsal and Twist are present in limiting concentrations (*Liberman et al., 2009*; *Rushlow et al., 1989*). Strikingly, despite the differences in predicted binding affinities and relative positions of the motifs, the elimination of any site individually had similar quantitative effects, reducing gene expression to approximately 60% of the peak wild-type level (*Figure 1C*, *Figure 1—figure supplement 1*, and *Supplementary file 2*). In contrast to this uniform picture, the impact of mutation of combinations of two Dorsal or Twist binding sites was highly variable, ranging from slightly lower expression to almost complete loss of activity (*Figure 1D*, *Figure 1—figure supplement 1*, and *Supplementary file 2*). Overall, the double activator site mutagenesis revealed a complex picture of the contributions of activator sites to gene expression. We hypothesize that the variable effects of different pairwise mutations, as opposed to the rather similar effects of individual site loss, indicates that there are multiple and distinct thresholds for specific biochemical events occurring on the enhancer. In contrast to the perturbation of Dorsal and Twist elements, removal of Snail repressor binding sites revealed stark differences in the significance of individual motifs for overall activity. Mutation of all four Snail sites caused pervasive expression in the mesoderm, as expected, while constructs with a single intact Snail2 or Snail3 site showed substantial but not complete repression (*Figure 1—figure supplement 1* and *Supplementary file 2*). Snail1 and Snail4 motifs were not nearly as effective at mediating repression, although these have similar binding affinities (*Figure 1E*, *Figure 1—figure supplement 1*, and *Supplementary file 2*). Snail4 and Twist2 sites overlap, which may impair Snail binding and reduce repression efficiency.

### Thermodynamic modeling of *rho* enhancer perturbation dataset

Direct examination of the data described above showed that the inter-relationships among activator and repressor sites were complex, and mutant phenotypes were not simply additive. Such complexity is a familiar facet of *cis*-element mutagenesis studies (*Swanson et al., 2010*; *Evans et al., 2012*). To extract the non-intuitive, quantitative information about transcription factor function and

**Table 1.** Parameters in each model. Number of parameters in cooperativity and quenching model combinations. 3 scaling factor parameters for Dorsal, Twist, and Snail are included in all models. Column 1 contains the nomenclature and number of parameters (in parentheses) for all 15 cooperativity models. Columns 2–9 in Row 2 contain nomenclature and number of parameters (in parentheses) for 8 quenching models. Each cooperativity and quenching scheme is described in the materials and methods section. The parameters being fitted for continuous functions are clearly laid out in this section. The following example illustrates the parameters in binned cooperativity and quenching functions. Model C14Q5: Parameters 1–3 are scaling factors for Dorsal, Twist, and Snail respectively. Parameters 4–15 reflect cooperativity (separate parameters existing for each type of protein-protein interaction)– parameters 4–6 represent Dorsal-Dorsal cooperativity (at 1–60 bp, 61–120 bp, and 121+ bp between bound Dorsal proteins), likewise parameters 7–9 represent Twist-Twist cooperativity, parameters 10–12 represent Dorsal-Twist cooperativity, and parameters 13–15 represent Snail-Snail cooperativity. Paramters 16–23 reflect quenching – parameters 16–19 represent Snail quenching Dorsal (at 1–25 bp, 26–50 bp, 51–75 bp, and 76+ bp between bound proteins), likewise parameters 20–23 represent Snail quenching Twist.

| | Quenching Model | | | | | | | |
|---|---|---|---|---|---|---|---|---|
| Cooperativity Model | Q1 – MSB (0) | Q2 – Linear (2) | Q3 – Logistic (2) | Q4 – Gaussian (2) | Q5 – Binned 4_25 (8) | Q6 – Binned 4_35 (8) | Q7 – Binned 4_45 (8) | Q8 – Binned 10_10 (20) |
| C1 – Linear (4) | 7 | 9 | 9 | 9 | 15 | 15 | 15 | 27 |
| C2 – Logistic (4) | 7 | 9 | 9 | 9 | 15 | 15 | 15 | 27 |
| C3 – Gaussian (4) | 7 | 9 | 9 | 9 | 15 | 15 | 15 | 27 |
| C4 – Binned 2_25 (4) | 7 | 9 | 9 | 9 | 15 | 15 | 15 | 27 |
| C5 – Binned 2_50 (4) | 7 | 9 | 9 | 9 | 15 | 15 | 15 | 27 |
| C6 – Binned 2_75 (4) | 7 | 9 | 9 | 9 | 15 | 15 | 15 | 27 |
| C7 – Binned 3_50 (6) | 9 | 11 | 11 | 11 | 17 | 17 | 17 | 29 |
| C8 – Binned 3_60 (6) | 9 | 11 | 11 | 11 | 17 | 17 | 17 | 29 |
| C9 – Binned 3_70 (6) | 9 | 11 | 11 | 11 | 17 | 17 | 17 | 29 |
| C10 – Protein Binned 2_25 (8) | 11 | 13 | 13 | 13 | 19 | 19 | 19 | 31 |
| C11 – Protein Binned 2_50 (8) | 11 | 13 | 13 | 13 | 19 | 19 | 19 | 31 |
| C12 – Protein Binned 2_75 (8) | 11 | 13 | 13 | 13 | 19 | 19 | 19 | 31 |
| C13 – Protein Binned 3_50 (12) | 15 | 17 | 17 | 17 | 23 | 23 | 23 | 35 |
| C14 – Protein Binned 3_60 (12) | 15 | 17 | 17 | 17 | 23 | 23 | 23 | 35 |
| C15 – Protein Binned 3_70 (12) | 15 | 17 | 17 | 17 | 23 | 23 | 23 | 35 |

interactions from these results, we created an extensive set of quantitative models. Based on the likely importance of cooperativity (*Kazemian et al., 2013*), we tested models that incorporated a variety of conceptions of distance-dependent homotypic and heterotypic cooperativity, as well as different distance-dependent 'quenching' or repressive interactions between repressors and activators (*Veitia, 2003*; *Wagner, 1999*; *Kulkarni and Arnosti, 2005*; *Arnosti et al., 1996*; *Barolo and Levine, 1997*). We systematically tested continuous and step functions to determine possible distance relationships affecting interactions. In all, 15 formulations for cooperativity and eight formulations for quenching were employed (*Table 1*). Combining these two types of formulations resulted in 120 different models, which were trained on the quantitative expression data of the 38 enhancer variants. Parameters were estimated using CMA-ES, a global genetic algorithm, and overall performance was calculated from the fit to all constructs, using root mean square error (RMSE) as the objective function (*Hansen et al., 2003*). Examination of model performance at three levels – global

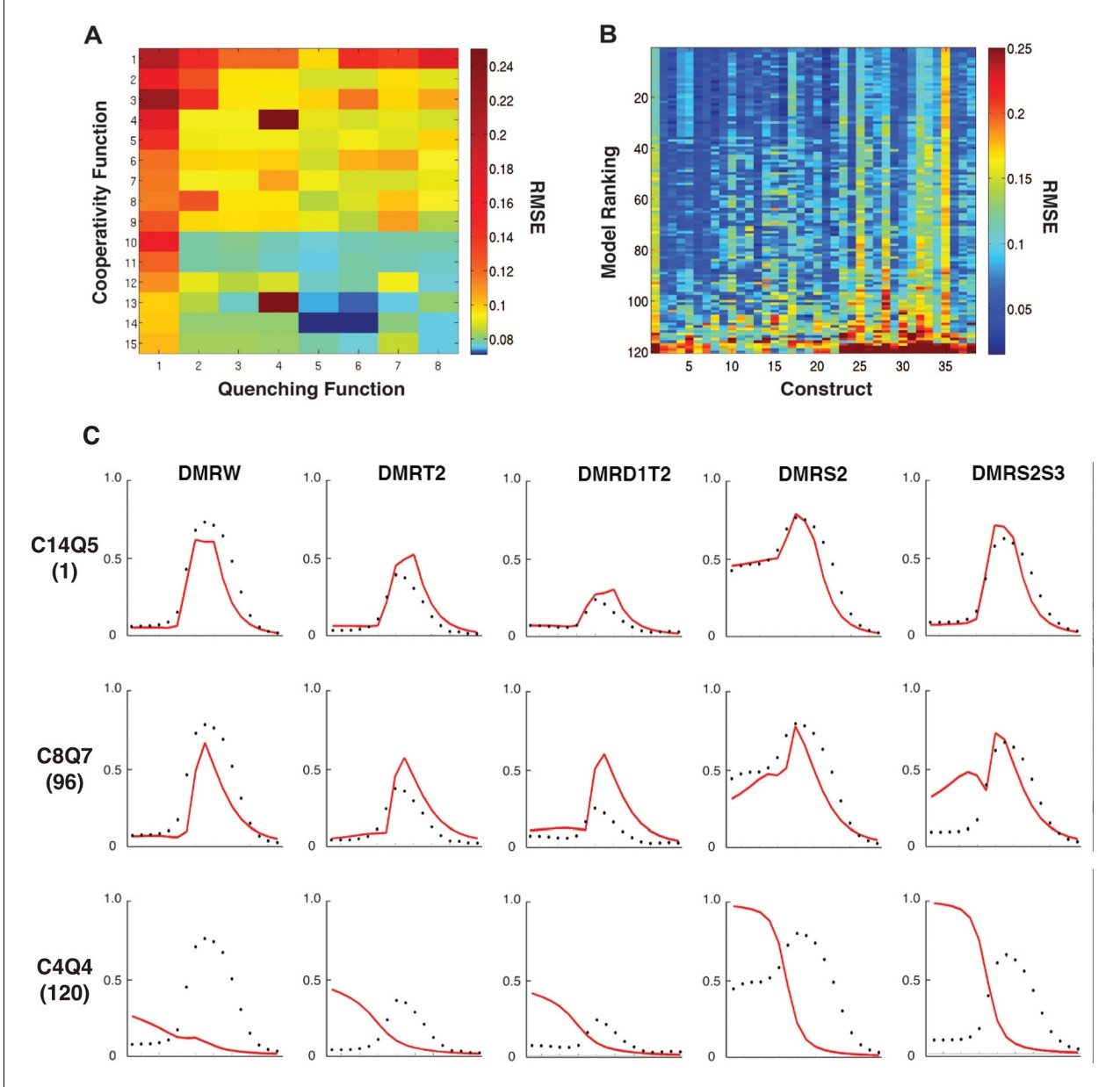

**Figure 2.** Performance of 120 thermodynamic models. (**A**) Heat map representation of results of thermodynamic modeling of *rho* perturbation dataset; the performance of each model is represented by a rectangle. Models were globally fit using Root Mean Square Error (RMSE), scale shown at right. Each model employs a distinct repression (quenching) scheme represented on horizontal axis, and a protein-protein cooperativity scheme on vertical axis. (**B**) Construct-by-construct performance ranking of 120 thermodynamic models ranked top to bottom. Vertical features are indicative of specific constructs that showed overall better or worse fit. (**C**) Examples of fitting of individual constructs by models ranking high (C14Q5; 1st overall), medium (C8Q7; 96th overall), or low (C4Q4; 120th overall). Black points indicate measured average mRNA levels; red lines are model predictions. The high-ranking model correctly predicts both low expression (repression) in mesoderm and activation in neuroectoderm, medium model exhibits modest deviations in prediction of both repression and activation, and the low-ranked model exhibits particularly poor fits for repression activity.

The following figure supplement is available for figure 2:

**Figure supplement 1.** Cross validation of thermodynamic models.

RMSE, construct-by-construct RMSE, and specific portions of the expression patterns - provided complementary insights into the nature of how the training data are fit, and the potential utility of the models on other enhancer sequences.

At a global level, the performance of the models was clearly co-dependent on both the cooperativity and repression formulations (*Figure 2A*). Overall cooperativity formulation performance varied according to the type of quenching formulation; the best quenching scheme for one cooperativity scheme was not necessarily optimal for other cooperativity schemes. Such interactive effects are likely a reflection of parameter compensation. Models with more parameters tended to outperform those with the fewest, as expected, but it was notable that this was not a strict correlation; the models with the most parameters were not as effective as those with fewer. Additionally, there were measureable differences between models with identical numbers of parameters, suggesting that the different formulation of the schemes was interrogating aspects of enhancer grammar critical for the *rho* enhancer variants we were fitting. Best overall fits were observed using a model with cooperativity values parameterized in three 'bins' of 60 bp (scheme C14) and quenching in four small 25 or 35 bp bins (schemes Q5 and Q6).

To examine construct-specific performance, we plotted a heat map illustrating individual construct fits for all 120 models (*Figure 2B*). The models, represented by rows, were ranked from best to worst based on global performance, and individual fits for each of the 38 *rho* enhancer variants were plotted in columns. The higher ranked models (blue rows) have generally lower RMSEs across all constructs, and groups of models with similar structures had similar patterns of better or worse performance on particular constructs. Some constructs were fit generally less well by most models. Those constructs containing the wild-type ensemble of activators (columns 1, 25, 28, 32) were among those that were less well fit than the bulk of constructs, from which we had removed activator sites; apparently the generally narrower neuroectodermal expression pattern of those constructs with some activator sites removed drove overall parameter fitting. Individual plots of the measured and predicted activity of individual enhancer modules show that some error arises from the models underestimating the activator potential specifically in more dorsal regions of the embryo, where activator concentration is limiting (*Figure 2C*; e.g. construct 1: DMRW). The lower quality fit for these constructs does not represent a model failure; rather, these constructs are likely to be especially informative for activator function, and a modeling effort that entirely lacks these types of constructs would be more over-fit and less informative than the present one. Further examination of individual plots for specific genes provides additional insight into the nature of which features influence RMSE scores the most (*Figure 2C*). A top-ranking model (C14Q5) accurately captures high and low levels of Snail repression, deviating only in underestimating the expression of *rho* in regions most limiting for Dorsal and Twist. An intermediate-scoring model (C8Q7) was partially successful in capturing general trends of activation with occasional overestimation of activity. This model misestimated Snail repression in some cases as well. The lowest performing model (C4Q4) suffered from poor estimation in Dorsal/Twist activity, as well as a general absence of repression by Snail.

Thermodynamic models calculate protein occupancy on an enhancer using descriptions of binding preferences distilled from position weight matrices (PWMs), which can incorporate in vitro or in vivo protein-DNA interaction data. A number of studies have tested DNA-binding preferences of Dorsal, Twist, and Snail, therefore we tested the effects of different PWMs on model performance. Three similar but non-identical PWMs were tested for each factor, and all possible combinations of these PWMs were utilized in refitting the data set with a selection of 24 of the entire set of models, including most of the top performers. Overall ranks of model performance remained similar, although RMSEs were lowest using the PWM of Snail that was employed in the global analysis in *Figure 2*; this PWM predicted all four sites previously identified by in vitro footprinting (*Figure 3*). Thus, the biophysical information about protein-DNA interactions for the transcription factors in this system appears to be sufficiently complete to provide a robust platform for thermodynamic modeling.

In addition to information about biophysical parameters of the system, model performance can be dramatically influenced by the nature of the perturbations studied. We used cross-validation, a statistical technique in which segments of the data set are left out during model fitting, to reveal possible overfitting, and determine the impact of certain portions of the data set on overall success in fitting. We performed cross-validation for the 24 models studied above both by leaving out specific parts of the dataset with common attributes (e.g. all Dorsal or Snail site mutations) as well as by randomly removing portions of the dataset. We used these two complementary approaches to not

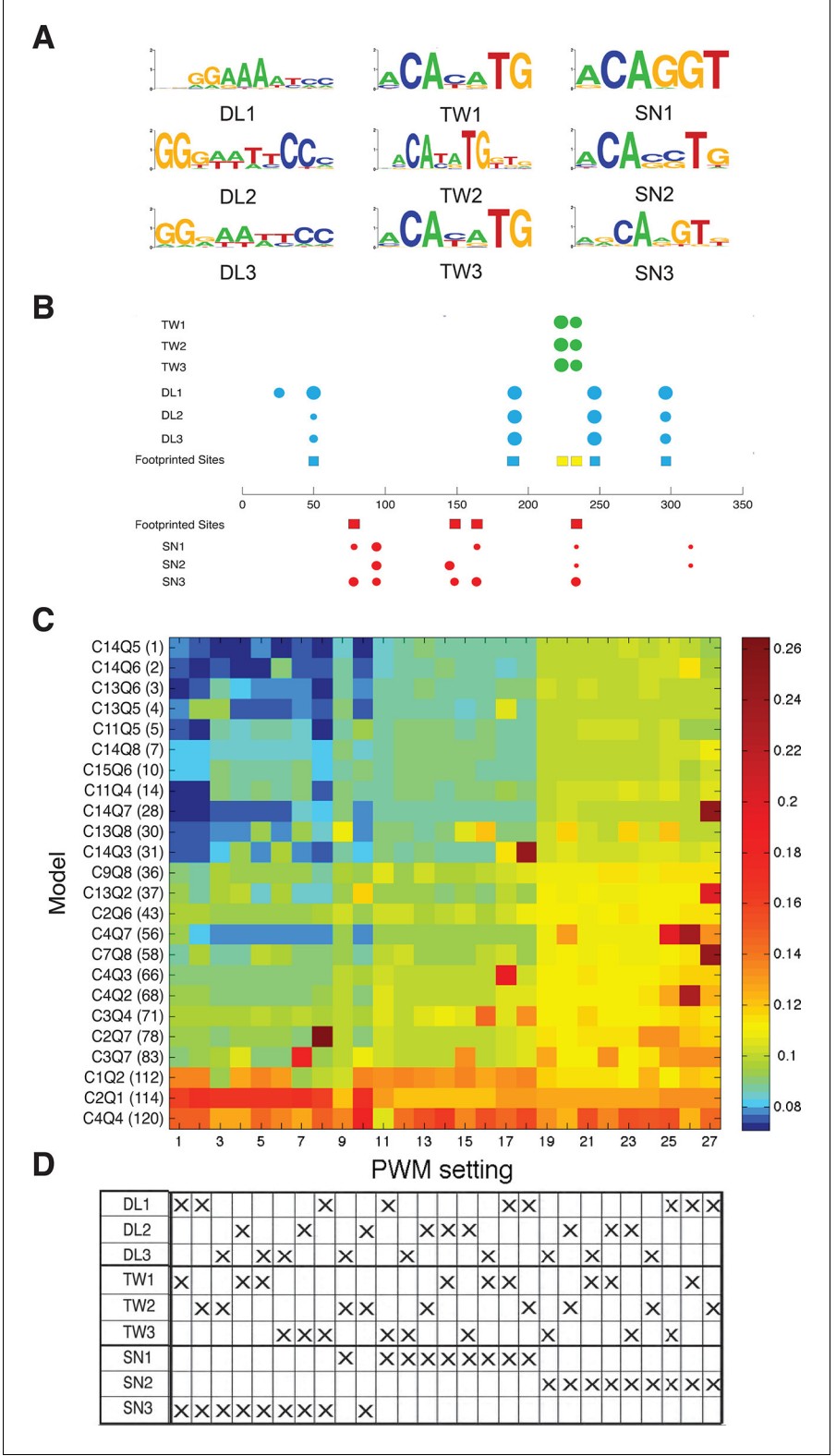

**Figure 3.** Effect of PWM settings on model performance. (**A**) Three versions of each binding site were considered as discussed in Materials and methods, drawing from previously reported in vitro or in vivo binding studies. (**B**) The correspondence between previously identified bindings sites from DNAseI footprinting studies (colored squares) and predicted binding sites using the diverse PWMs (circles) is superimposed on the structure of the *rho* enhancer. The size of the circles indicates the significance of the match using the particular PWM for that factor; *Figure 3 continued on next page*

*Figure 3 continued*

Twist sites in green, Dorsal in blue, and Snail in red. (C) Heatmap showing global performance of 24 selected models using 27 different PWM combinations for Dorsal, Twist and Snail. Results were clustered by performance; columns of PWM settings were arranged with lowest average RMSE values to the left, then rows of models were arranged with lowest global RMSE towards the top. (D) PWMs used for different settings are denoted by 'X's; Snail3 PWM (used in the rest of this study) shows strong correlation with best model fitting.

only test how much data is needed for the model fitting to be successful, but also to emphasize how important experimental design is in a study such as this one. Some studies rely on rely on saturation mutagenesis to achieve the same effect, but testing thousands of variants is impractical in this system, so a targeted survey was called for (*White et al., 2013*). As anticipated, models were in general less able to fit constructs that were left out of the dataset, indicating the contribution that these particular enhancer constructs made to the fitting (*Figure 2—figure supplement 1*). Despite similar performance on the entire dataset, individual models showed differing levels of sensitivity to changes in the scope of the data set used for model fitting. For instance, RMSE values for C13Q5 were modestly increased by either systematic or random leave-sets-out treatment, while C11Q4 was more dramatically impacted. In general, removal of entire classes of constructs had a more profound impact than removal of randomly selected constructs.

To better understand how specific constructs contribute to model performance, we analyzed how leaving out certain sets of constructs affects model predictions on each individual construct, including those used for fitting and those left out (*Figure 2—figure supplement 1*). The most profound effects were seen with the omission of the Snail repressor site constructs, where almost all models had striking increases in RMSE for all Snail constructs. Evidently, the elimination of activator sites alone is not sufficient to provide insights into how Snail affects the enhancer's expression. Collectively, the scope of mutations assayed, targeting recognized Twist, Dorsal, and Snail sites, was sufficient to provide the required information to the modeling effort. The different effects of activator and repressor site mutations illustrate the importance of perturbation of each of these elements to fully explore the functional terrain of the enhancer; for a small number of constructs, a random perturbation of *cis*-regulatory sequences may not uncover the most informative changes related to transcriptional relationships (*Patwardhan et al., 2012*; *Sharon et al., 2012*; *Smith et al., 2013*).

All thermodynamic models used here characterized transcription factor activity by estimating parameters for activation and repression potential, as well as the distance-dependence of cooperativity and repression. Several broad trends emerged from analysis of parameter values across many model types, revealing possible biological implications for the modeled enhancer (*Figure 4*, *Figure 4—figure supplement 1*, *Supplementary file 3*). First, overall model performance was improved by specifying separate parameters for activator and repressor cooperativity (*Figure 2A*; C1-9 vs. C10-C15). In these latter models, repressor-repressor cooperativities were consistently small, implying that different Snail sites did not show more than additive contributions, a finding that is consistent with the induction of localized chromatin compaction by this class of short-range repressor, which may interfere with simultaneous interactions by repressors (*Li and Arnosti, 2011*). Activator-activator cooperativities were in contrast uniformly high, and in general did not show sharp distance dependencies. Such interactions may represent indirect interactions among activators, and may reflect joint repulsion of nucleosomes, or cooperative attraction of coactivators important for engaging the transcriptional machinery. The loose cooperative interactions of activators would explain the need for clustering of these proteins' binding sites without strong constraints on their exact positioning, a flexibility that is seen with many developmental enhancers (*Figure 4B*; *Arnosti and Kulkarni, 2005*). This long-distance cooperativity contrasts with the short-range interactions often included in thermodynamic descriptions of transcription factor cooperativity (*Segal et al., 2008*). A second trend noted in many but not all models was the distance-dependence of transcriptional repression, a feature inferred from previous studies of the Knirps, Snail, and Giant short-range repressors (*Gray et al., 1994*; *Arnosti et al., 1996*; *Hewitt et al., 1999*); (*Figure 4—figure supplement 1B*; *Supplementary file 3*). Thus, the overall landscape of estimated parameters both reflects certain known aspects of repression, and sheds light on activator cooperativity, which may involve primarily indirect, not rigidly constrained interactions in this system.

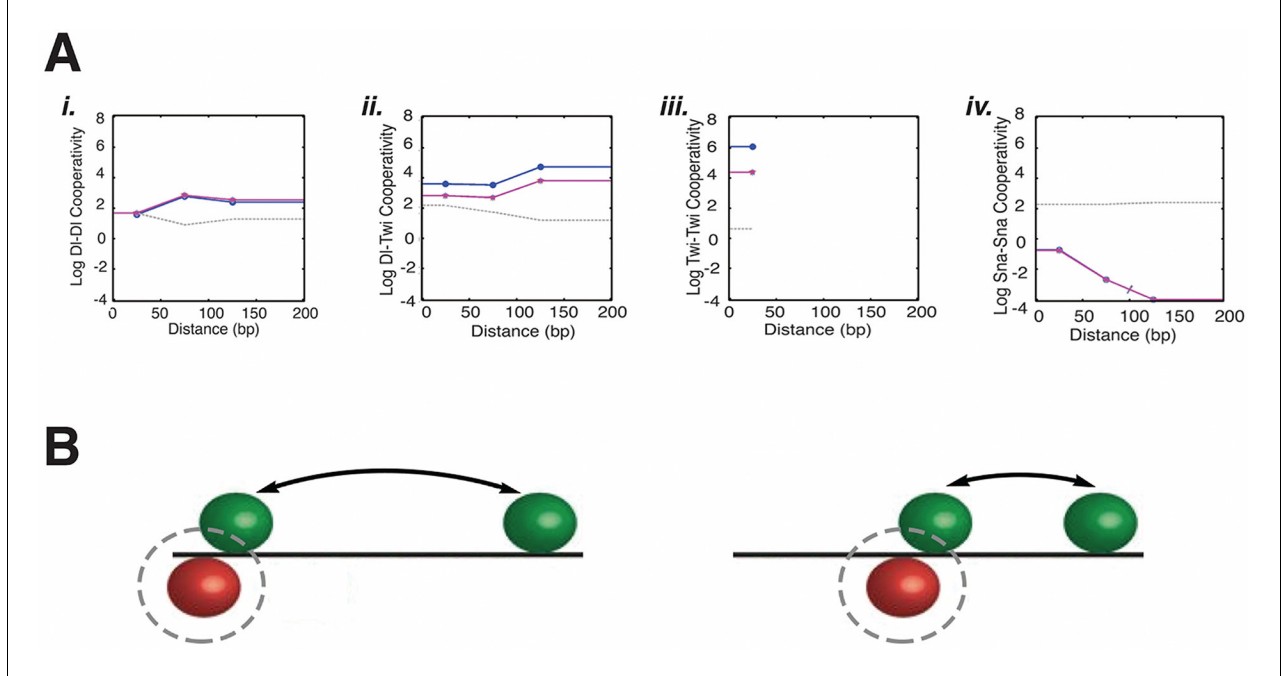

**Figure 4.** Estimated model parameters suggest biological properties of long-distance activator-activator cooperativity and weak repressor cooperativity. (**A**) Representative estimated cooperativity parameters for Dorsal and Twist transcriptional activators show a trend of high levels of cooperativity with little distance dependence, consistent with indirect, long-range effects i. Dorsal-Dorsal, ii. Dorsal-Twist, iii. Twist-Twist, and iv. Snail-Snail cooperativity values of model C13Q3. Low values estimated for repressor cooperativity are consistent with weak interactions between Snail proteins. One of the five runs for fitting this model had significantly higher RMSE than the other four; the corresponding parameters are indicated by a gray line. Other runs produced similar RMSE and parameters (magenta and blue). (**B**) Illustration of how global long-range cooperative interactions between activators permit relaxed constraints on binding site arrangement, while distance-dependent repressor positioning anchors Snail sites to activator sites.

The following source data and figure supplements are available for figure 4:

**Figure supplement 1.** Global trends observed for parameter values and evidence of compensation.

**Figure supplement 2.** Model sensitivity analysis.

**Figure supplement 2—source data 1.** Sensitivity analysis.

As we noted in previous studies, estimated parameters from thermodynamic models show different levels of sensitivity due to the structure of models and of data (*Dresch et al., 2010*). We tested the set of 24 models with synthetic data sets to determine sensitivities, and determined that most parameters showed extensive mutual dependence and compensation, evidenced by the low first-order and high second-order sensitivities (*Figure 4—figure supplements 1* and *2*). The observed convergence of cooperativity and quenching parameters for a number of models during our data fitting thus suggests that although first-order sensitivities are low, there is important information inherent in these values.

The objective of our study was to develop a quantitative understanding of novel enhancer sequences, using our models in a predictive mode. To test their efficacy, we first asked whether the perturbation analysis and modeling of *rho* was capable of uncovering a key property of modular enhancers in the embryo, namely the short range of transcriptional repressors that prevents 'crosstalk' with other nearby elements (*Gray et al., 1994*; *Courey and Jia, 2001*; *Payankaulam et al., 2010*). Previous work showed that Snail protein is representative of this major functional class of repressors, in studies that involved moving Snail binding sites within transcriptional regulatory regions (*Gray et al., 1994*). Our perturbation dataset did not involve moving Snail binding sites around, however, the top third of models in our test set were able to accurately capture the effects

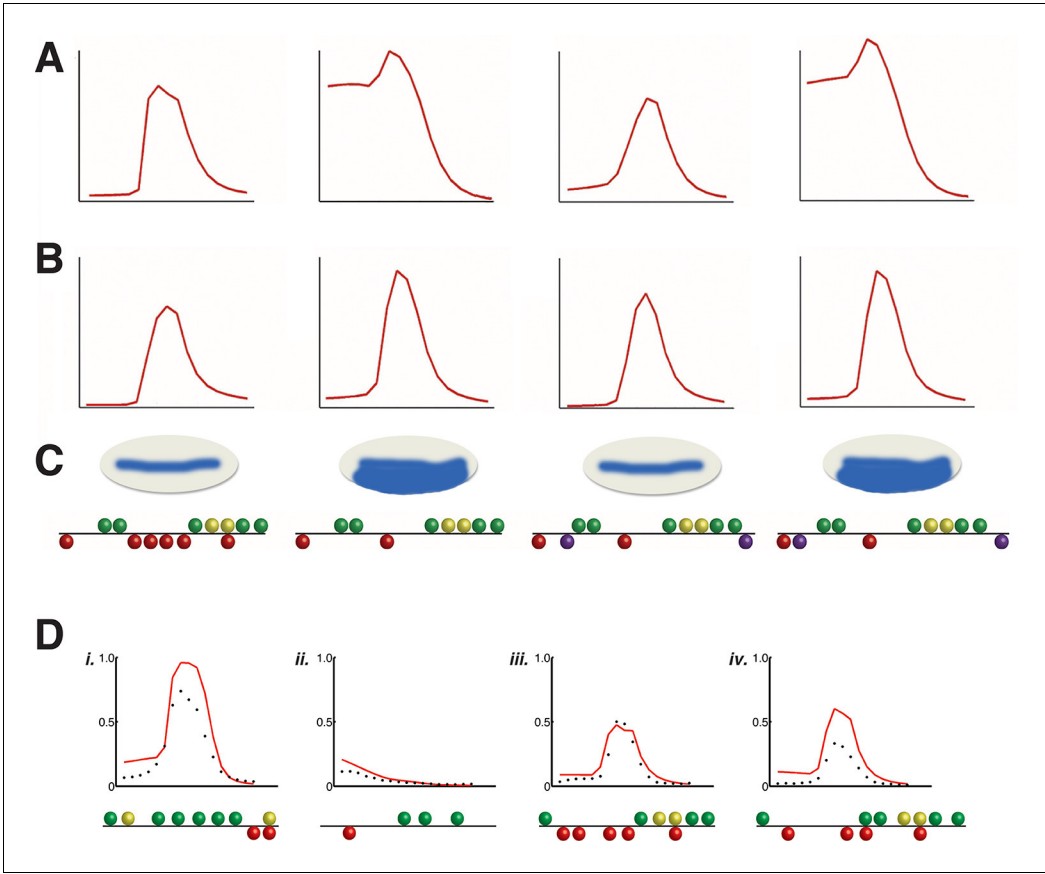

**Figure 5.** Application of *rho cis*-regulatory 'grammar' to heterologous regulatory elements. Discovery of short-range repression from modeling of *rho* perturbation dataset. (**A**) High ranking model C13Q6 (3rd overall) correctly predicts distance-dependent significance of Snail sites within enhancer element; specifically, the third but not fourth construct positioned ectopic Snail binding sites (purple balls) close enough to Dorsal 1 and Dorsal 4 motifs to repress the expression in ventral-most parts of the embryo. (**B**) lower ranking model C2Q7 (78th overall) correctly predicts activation potential, but fails to detect distance effects impacting repression by Snail. (**C**) Shows cartoons of the qualitative expression found by Gray and colleagues and the reporter genes tested in embryos (*Gray et al., 1994*). (**D**) Correctly predicted expression patterns of *cis*-regulatory elements i. *D. melanogaster vnd* enhancer, ii. *D. melanogaster twi* proximal element, iii. *D. erecta rho* enhancer, and iv. *D. ananassae rho* enhancer. X-axis denotes distance from ventral end of the embryo, and Y-axis denotes expression values. Predictions of model C14Q5 (ranked 1st overall) are shown as red lines; measured average transgene expression in black points.

of loss of Snail sites from a 700 bp native *rho* element, as well as correctly characterize modified enhancers in which novel Snail sites were inserted into active as well as inactive locations (*Figure 5A–C*). To determine whether modeling from *rho* can be broadly extended to other regulatory elements, we measured the quantitative output of other enhancers targeted by Dorsal and Twist, and compared their expression patterns with the predictions of a spectrum of our models. Heterologous elements from *D. melanogaster* and homologous *rho* enhancers from other *Drosophila* species were successfully modeled by the models trained on the *rho* dataset (*Figure 5D* and *Figure 6*).

The previous assays tested the models on elements already known to act as Dorsal response elements and to bind Dorsal/Twist. To determine whether models would be able to identify and correctly quantitatively score enhancers embedded in general genomic sequences, we tested the panel of models on 30–40 kb regions flanking a number of developmental genes. The regions were tiled in 500 bp overlapping sequences and predictions generated with parameters from the *rho* dataset fitting. Analysis of the neuroectodermal *brk* and *sog* genes, as well as the more widely expressed *twi* gene showed that most intergenic sequences did not give rise to any appreciable predicted

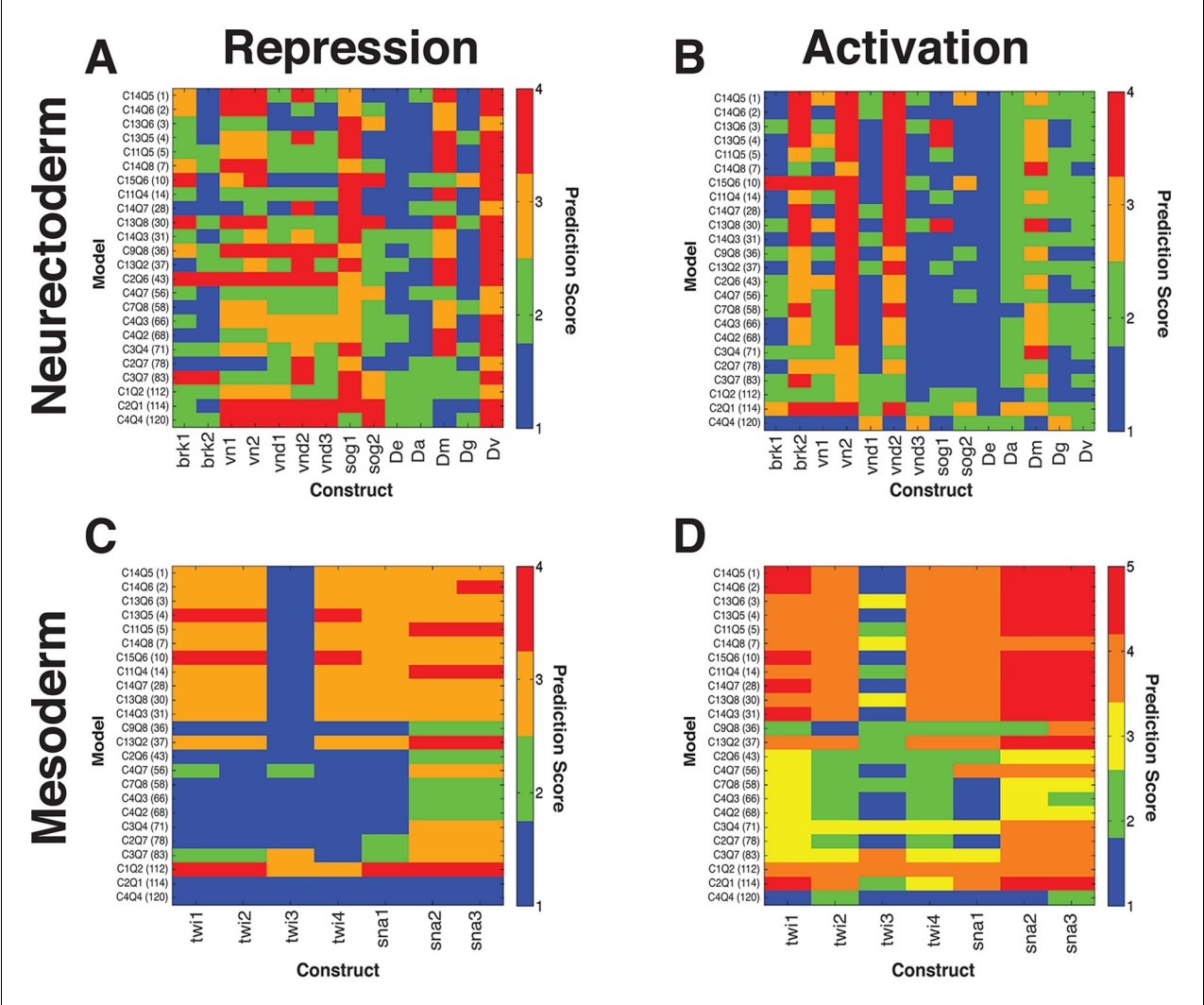

**Figure 6.** Thermodynamic models predict quantitative outputs of additional, non-*rho* enhancers in *D. melanogaster* and putative *rho* enhancers from related drosophilids. (**A**, **B**) neuroectodermal elements scored for repression by Snail and activation by Dorsal and Twist, respectively. (**C**, **D**) mesodermal elements scored for lack of repression by Snail and preferential activation in mesoderm, respectively. Lowest ranked models performed poorly for correct prediction of Snail activity on neuroectodermal enhancers, but were also not liable to make a false positive call on Snail activity for mesodermal elements (**A**, **C**). The *D. erecta* rho element was correctly called by most models, while some *sog, vn, vnd*, and *brk* constructs were less well fit by most models, indicating that there are likely enhancer-specific features of these elements that are not sampled by *rho* variations. For mesodermal elements derived from *snail* and *twist* proximal regions, those models that correctly scored Snail activity as low were able to produce more 'mesoderm-specific' type patterns. No model scored consistently highly across all constructs, but many performed well in a complementary fashion, suggesting that individual models are not overfit in identical manners. Activity of regulatory elements was experimentally measured in transgenic *D. melanogaster*, and output compared with the predicted expression from 24 models thermodynamic models fit on *D. melanogaster* rho expression. Fitting for predicted vs. measured expression was assessed using quantitative measures for correct expression as described in Materials and Methods. Lower scores represent better fits. *D.erecta, D.ananassae, D.mojavensis, D.grimshawi,* and *D.virilis* putative *rho* elements were assayed (sequences indicated in ***Supplementary file 1***).

expression (***Figure 7*** and ***Figure 7—figure supplement 1***). To help one visualize how well the models agreed (or disagreed) on particular regions, average predictions from all 24 models were also generated for each 500 bp window, and plotted along with the standard deviation above and below the mean. A strong consensus among the 24 models for activity was found in two regions flanking the *brk* gene; these regions include mapped embryonic enhancers (primary and shadow) that are known to bind Dorsal and Twist in vivo (***Figure 7A***). Some regions were less uniformly predicted by the models, and in these cases there was less uniformity about the prediction for mesodermal or

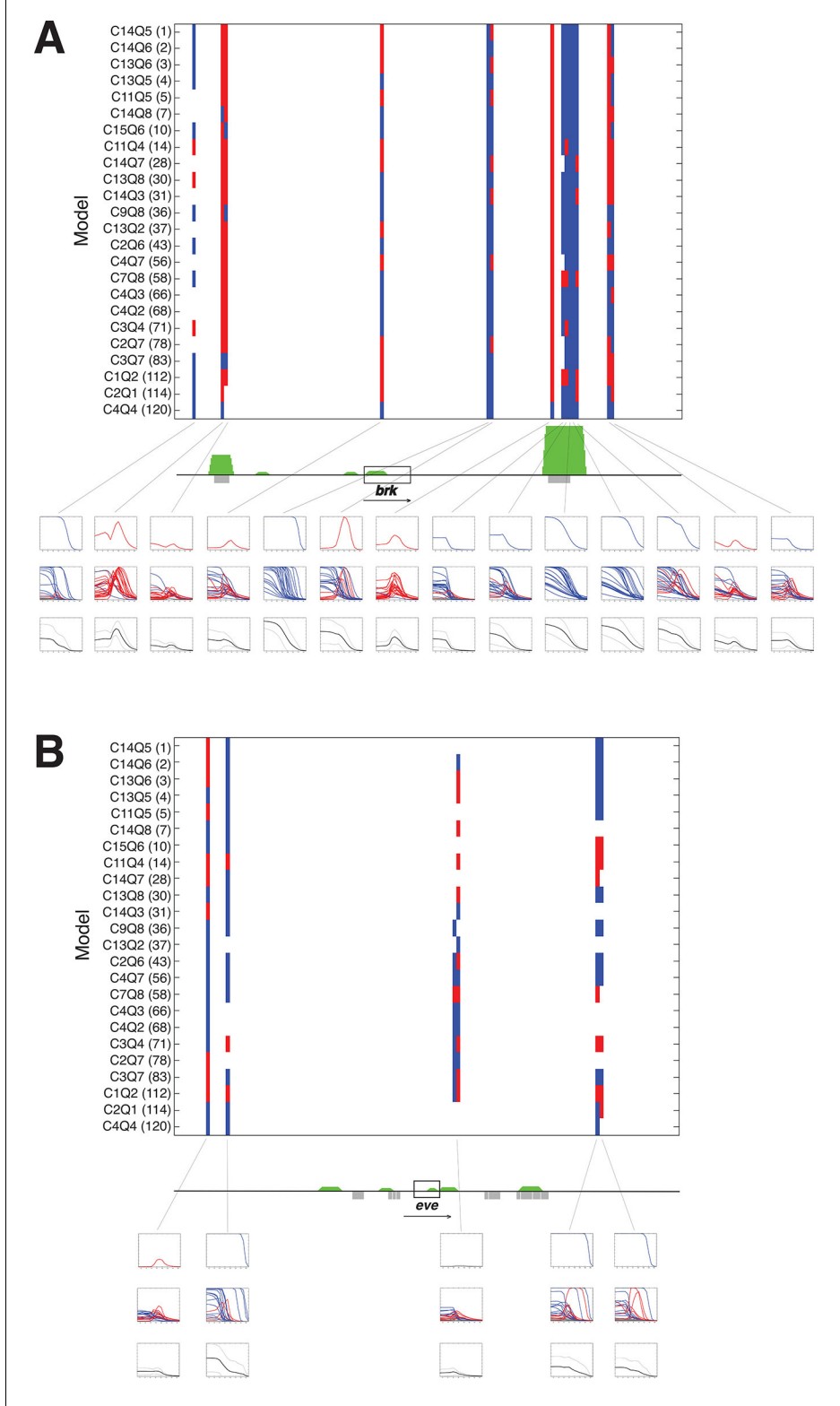

**Figure 7.** A panel of models fit on *rho* enhancer predicts additional neuroectodermal enhancer elements. (**A**) Predicted expression activity of a genomic region (36 kb) flanking the neurectodermal *brk* gene, using a panel of 24 models, including 21 higher and three lower performing models. Blue marks indicate predicted mesodermal pattern, red neuroectodermal. The transcribed area for *brk* is indicated by the black rectangle; green peaks indicate in vivo binding for Dorsal, Twist, and Snail; previously mapped embryonic enhancers are indicated by gray

*Figure 7 continued on next page*

*Figure 7 continued*

rectangles. Rows of plots below the gene show expression patterns of: first row – top model, second row – composite of all 24 model predictions, third row – average of all 24 model predictions in solid black with standard deviation from the mean shown above and below in dotted black lines. Note that the average plot is similar to a plot of a weighted average using RMSEs or AICs to weight each model's output. (B) Similar scan of the *even-skipped* locus (32 kb), which is not regulated by Dorsal, and where few areas are consistently identified as potential Dorsal/Twist/Snail enhancers.

The following figure supplement is available for figure 7:

**Figure supplement 1.** A panel of models fit on *rho* enhancer predicts additional neuroectodermal enhancer elements around *sog* and *twi*, with lower scores around *ftz* control region.

neuroectodermal expression, indicating that Snail activity was variably predicted. These predictions tended to agree well with genomic protein occupancy information, although there were some regions predicted as enhancers where there was no protein occupancy measured. In contrast, there were fewer predicted active elements in the region surrounding *even-skipped (eve)* locus, which is expressed in an anterior-posterior pattern independent of Dorsal/Twist. In addition, for intergenic regions flanking *eve*, there was greater variability among models in predicted expression levels, patterns of expression, and whether an element was even likely to be active (*Figure 7B*). Similar results were noted for neuroectodermal and mesodermal-specific *short gastrulation (sog)* and *twist (twi)* genes respectively, as well as the non-target gene *fushi tarazu (ftz)* (*Figure 7—figure supplement 1*). The use of multiple models as a predictive 'jury' may help overcome the overfitting that is inherent in modeling, as the effects of such bias for a given model may be more or less prominent in different contexts. The ensemble approach of averaging the 24 model predictions provides a convenient metric for transcriptional potential predictions, yet the graphic display of the individual models also serves an important role, to highlight particular regions where there is a high degree of uniformity about the prediction (e.g. primary and shadow *brk* enhancers) and those in which the models diverge sharply (*eve* proximal region).

## Discussion

Bioinformatic sequence analysis and direct comparison of chromatin features have been used to identify cis regulatory elements in metazoan genomes, however the significance of patterns of binding sites within such elements remains obscure. Many models simply rely on the number of sites to infer some level of activity (*Kazemian et al., 2010*; *Heintzman et al., 2009*; *Bonn et al., 2012*). Our study indicates that the arrangement and quality of transcription factor binding sites contains an essential common 'grammar' of *cis*-regulatory regions that is shared across distinct regulatory elements; this stands in contrast to recent studies suggesting that in some cases transcription factors may recognize their target sequences primarily through protein-protein rather than protein-DNA interactions (*Kulkarni and Arnosti, 2005*; *Junion et al., 2012*). Our models were biologically constrained, leading to structural similarities between the different model formulations. Although this may contribute to the clustering of parameter values we have observed, given the overall quality of predictions across the genome, we believe these models and parameter values to be biologically informative. Predictions of enhancer activity from sequence alone remain a challenge, but our study clearly shows that with a suitable training set, there are identifiable aspects of enhancer composition that allow for functional insights of mesodermal and neuroectodermal genes of the Dorsal regulon. Models such as these can guide future studies that mine population- and species-level variation of genomic sequences (*Mackay, 2010*).

Two features were important for the successful implementation of this thermodynamic modeling: development of a high quality perturbation dataset, which in our case involved several dozen carefully designed constructs whose transcriptional output was measured quantitatively, and exploration of a variety of model structures. As we determined when applying these models to genome discovery, using a panel of models provides a more robust platform for interpreting possible regulatory information; the particular parameter sets developed from fitting the *rho* perturbation data may represent a certain level of overfitting with respect to all potential Dorsal/Twist target enhancers. These

errors are partially cancelled out when the predictions of several dozen models can be evaluated, providing in effect a type of 'weather report' showing a certain chance of regulation at each element. Here, we have also used an ensemble approach to combine these predictions, averaging model predictions; an approach similar to that which was implemented in a recent study by Samee et. al. (*Samee et al., 2015*). However, in their work they use a single thermodynamic model formulation and average the predictions obtained from different parameter sets with similar fits to their perturbation data.

In addition to recapitulating known facets of *cis*-regulatory grammar in *Drosophila* including the distance-dependent activity of repressors, this modeling approach identified critical quantitative features of activator and repressor interactions at enhancers. Specifically, for Dorsal and Twist, key regulators of dorsal-ventral polarity in the early embryo, our modeling predicts strong distance-independent homo- and heterotypic cooperativity between activators, but weak cooperativity between the Snail short-range repressors. These effects are likely linked to chromatin-based activities of these proteins (*Li and Arnosti, 2011*). Previous, simpler models suggest that direct protein-protein interactions within a range of ~ 50 bp are likely to dominate such functional interactions (*Segal et al., 2008*). The extensive survey of model structure allowed us to reject assumptions this simplifying assumption. These predicted properties of transcription factor activity were robustly observed for differently structured models, making us confident that they are biologically relevant and not the result of overfitting, and the initial application of models to genomic sequences points to the future use of these models in genomic characterization of the Dorsal-Twist regulon. Our ability to gain insight into the transcriptional landscape of this network is a proof of concept demonstrating the utility of such a modeling approach for identifying regulatory relationships in other systems.

Our modeling approach was focused on three main regulators of *rho* activity, and did not take into account additional factors affecting *cis*-regulatory elements, such as intrinsic chromatin occupancy, modification of transcription factor activity by signaling, or additional proteins involved in regulation of some of the enhancers. Nonetheless, future models will incorporate such additional layers of information in more complex treatments, such as including basal chromatin patterns to bias the accessibility of transcription factor sites (*Bryant et al., 2008*; *Floer et al., 2010*). Our study has addressed only a fraction of the diversity of factors present in the fly embryo; we deliberately focused on the early Dorsal regulon for the richness of quantitative resources available to it, including transcription factor concentrations, binding specificity of trans-acting factors, and genomic data on in vivo targets. With continuing advances in genomics and high-throughput technologies, quantitative modeling can extend our knowledge of other regulons, building on extensive descriptions of gene and protein expression, genomic protein occupancy, and genomic variation. A greater promise for modeling the quantitative grammar of genomes will be in 'personalized genomics', in which investigators predict the effects of sequence variation in human populations, and their physiological relevance in development and disease (*Corradin and Scacheri, 2014*; *Newman and Young, 2010*).

## Materials and methods

### Reporter gene constructs

The 318-bp *rhomboid* neurectodermal enhancers were cloned into *Age*I and *Fse*I restriction sites of the pHonda1 pattB-based targeted integration vector (*Ip et al., 1992*; *Sayal et al., 2011*). Enhancers were assembled from 40–60 bp overlapping synthetic 5'-phosphorylated oligonucleotides with 10 bp overhangs, which were annealed, and then ligated into pHonda1. The footprinted binding sites for Dorsal, Twist and Snail, as well as two predicted E-box motifs thought to be bound by bHLH factors, were mutated as follows using sequences previously shown to affect *rho* enhancer activity (*Ip et al., 1992*; *Jiang et al., 1991*):

    Dorsal1 - GGGAAAAACAC to TTTAAAAACAC
    Dorsal2 - CGGAATTTCCT to CGTCAGTTAAT
    Dorsal3 - GGGAAATTCCC to TCTAGATTATC
    Dorsal4 - GGGAAAGGCCA to AGGCCTGGTCA
    Twist1 - CGCATATGTT to ACGCGTTGTT*
    Twist2 - AGCACATGTT to ACGCGTTGTT
    Snail1 – CAACTTGCGG to CAGAGCTCGG

Snail2 - CACCTTGCTG to CAGGAGCTTG*
Snail3 – CCACTTGCGCT to CCGCCGGCGT*
Snail4 – GCACATGTTT to GCATATGTTT
bHLH1 - CATTTG to TGATTC*
bHLH2 - CAAGTG to TAGCGA*
(*novel mutations developed for this study)

The bHLH1 and bHLH2 sites were mutated simultaneously. The mutations are predicted to reduce the binding score for each transcription factor to near background values. Additional wild-type enhancers for other genes were created by PCR amplification from genomic DNA or by assembly using oligonucleotides as indicated above. We used the Clusterdraw bioinformatics tool to identify putative *rhomboid* regulatory sequences in non-*D. melanogaster* genomes (*Small et al., 1992*; *Papatsenko, 2007*). *Supplementary file 1* contains details of *rhomboid* and other genes' enhancer sequences and their nomenclature. All constructs were integrated into the same site on chromosome 2 (chromosomal location 51D; Bloomington stock center's stock #24483). DNA microinjections were performed in-house and by Rainbow Transgenic Flies, Inc. Transgenic lines were made homozygous, and only embryos from homozygous fly lines were used for confocal microscopy.

## Immunofluorescent in situ hybridization

Embryos were collected and fixed as previously described (*Small et al., 1992*; *Kosman et al., 2004*; *Janssens et al., 2005*). Immunofluorescent in situ hybridization was done essentially as previously described with some modifications (*Kosman et al., 2004*; *Janssens et al., 2005*). All washes were done in 1 ml volume. About 50 µl of fixed embryos stored at -20°C in methanol were briefly washed six times with 100% ethanol, followed by a wash in xylenes for 30 min, and lastly, six times again with 100% ethanol. The embryos were then washed four times with 50%methanol-50%phosphate buffer-0.1%-Tween 80 (PBT; 137 mM NaCl, 4.3 mM $Na_2HPO_4$, and 1.4 mM $NaH_2PO_4$) and then with PBT four times, each for 2 min with continuous rocking. Embryos were washed in (1:1, v/v ratio) PBT/ hybridization solution (hybridization solution: 50% formamide, 5X SSC [0.75M NaCl and 75 mM Na-citrate], 100 µg/mL sonicated salmon sperm DNA, 50 µg/mL heparin, and 0.1% Tween 80) for 10 min, and then briefly in hybridization solution for 2 min. New hybridization solution was added, and the tubes were placed for 1 hr in a water bath at 55°C. Previously titrated antisense RNA probes of digU-labeled *lacZ* and biotin-labeled *eve* and *sna* were heated in 65 µL hybridization solution at 80°C for 3 min and directly placed on ice for 1 min; hybridization solution was completely removed from the embryos, and the probes were added to the embryos in a final volume of 65 µL in each tube, and incubated at 55°C overnight. After incubation, 1 mL of 55°C hybridization solution was added to each tube; all tubes were rocked at room temperature for 1 min, hybridization solution was changed, and tubes were incubated for another 1 hr at 55°C, followed by four washes with hybridization solution for 15 min each at 55°C and with hybridization solution and PBT (1:1, v/v ratio) two times at room temperature for 15 min. Five more washes were done with PBT for 10 min with rocking at room temperature. The embryos were washed with a blocking solution consisting of a mixture of PBT and 10% casein in maleic acid buffer (Western Blocking Reagent; Roche, Indianapolis, IN 11921673001) (4:1, v/v ratio). 0.5 ml of a 4:1 v/v mixture of PBT and 10X blocking solution containing primary antibodies (3 µl of sheep anti-digoxigenin, (Roche 11333089001); 1 µl of mouse anti-biotin (Invitrogen 03–3700) was added, and the tubes were rocked at 4°C overnight. Embryos were washed four times each with PBT for 15 min at room temperature. 0.4 ml of mixture of PBT and 10% casein blocking reagent and PBT (4:1 v/v), containing 8.0 µl of each secondary antibody (donkey anti-sheep Alexa 555 (Invitrogen A-21436) for detection of *lacZ* mRNA and donkey anti-mouse Alexa 488 (Invitrogen A-21202) for detection of *eve* and *sna* mRNA) Secondary antibodies that had been pre-absorbed for at least 2 hr against fixed *yw* embryos in PBT and 0.4 µl of TOPRO-3 DNA dye (Invitrogen, T3605) were also added to each vial. Tubes were covered with aluminum foil to protect them from light and incubated overnight with rocking at 4°C. Embryos were then washed with PBT four times at room temperature for 5 min. with rocking, and washed in glycerol-PBT (7:3, v/v ratio) for 2 hr until the embryos settled to the bottom of the tubes. The embryos were then resuspended in 0.4 mL glycerol-PBT (9:1, v/v ratio) and 0.2 mL of Permafluor™ mounting medium (Fisher TA-030FM), mounted on labeled slides, and covered with large rectangular Corning cover slips (No. 1.5; 24 X 50 mm). The slides were protected from light and stored flat at room temperature until the embryos were imaged.

## Confocal laser scanning microscopy

An Olympus Spectral FluoView FV1000 Confocal Laser Scanning Microscope (Olympus, Center Valley, PA) configured on an IX81 inverted microscope was used for capturing the confocal fluorescent images. For each scan of mounted embryos on a particular day of imaging, the same microscope settings for wild-type *rho* transgenic embryos were employed to all images to allow for direct comparison of intensities. The 488 nm argon laser was used for excitation of the Alexa 488; the 559 nm solid-state laser was used for excitation of the Alexa 555, and the 635 nm solid-state laser was used for excitation of the TOPRO-3. Emitted fluorescence was detected using a 500–545 nm band pass filter for detection of the Alexa 488, a 570–625 nm band pass filter for detection of Alexa 555, and a 655-755 nm band pass filter for detection of TOPRO-3. The pinhole aperture was set to 1.0 Airy unit. PMT voltage was set at maximum for images obtained from embryos transgenic for the wild-type *rho*NEE enhancer, avoiding saturation of signal intensities. Other constructs were imaged subsequently on a single day without changing any of the microscope settings. Embryos were imaged at a scan speed of 12.9 s/scan and a Kalman average of 2. A total of 21–30 confocal images through the Z thickness were acquired for each embryo with a Z-step interval of 1.16 μm per step. CLSM image data was stored as three separate stacks and projections of images for each channel. The section dimensions were 333 mm in length and width, and 1.73 mm in depth. Fluorescence pixels were recorded as 12-bit images and stored as TIFF files. To control for fading of signal post-staining, *rho* constructs containing the wild-type ensemble of activators were stained in parallel and used to normalize overall signal intensity for each imaging day. Stained embryos were imaged within a week to minimize loss of signal. Differences in probe bleaching, laser intensity, gain settings of the CCD etc. all impact overall signal intensity. For the 348 control images captured over 53 imaging sessions, the range of average peak intensities, prior to any background subtraction or normalization, was 56.8 – 255 units (only 3 were at 255, saturation value). The mean was 138.0, with a standard deviation of 36.8. Thus, for the large majority of captured signals, the day-to-day differences in intensities were not very great, and normalization procedures were not changing values by large factors. The background signals from non-expressing portions of the embryos were 52.4 +/- 23.8 (S.D.), thus considerably (2X) below the signal; and in all cases the strong signals measured on any day were well above background measured on any day.

## Image processing

All confocal microscopy images were processed in a six-step procedure involving binary image generation, rotation, resizing, background subtraction, normalization and intensity-value extraction. Binary image generation, rotation and resizing were done as described previously (*Myasnikova et al., 2005*; *Ay et al., 2008*). The area of interest for all embryos comprised a region spanning from 40–60% egg length on the anterior-posterior axis. Ten samples, uniformly spaced, were taken from this region, plotted together and averaged along the dorsal-ventral axis (as is illustrated in *Figure 1*). For background subtraction, analysis of background signals from non-transgenic, *yw* flies showed a parabolic-shape (*Bergman et al., 2005*; *Myasnikova et al., 2005*), therefore a quadratic function was fit to the region of the signal representing the dorsal ectoderm, where *rho* is not expressed, and subtracted from the raw fluorescent signal. To normalize signals, values from each image were normalized to the average peak (>95%) wild-type signal obtained during the same imaging session. This procedure allows for images to be compared for a single construct imaged on multiple days, as well as to compare intensity from one construct to another. The average intensity profiles, along with standard error calculations are given in *Figure 1—figure supplement 1* and *Supplementary file 2*.

For model fitting, we discretized the continuous expression data. We did this by taking the data in the region from 0–40% of the DV axis (approximately 102 pixels in each image), and averaging every 6 pixels to result in 17 data points corresponding to this region; hence a data point every 2.5% of egg height (as is illustrated in *Figures 2* and *5*). We excluded the dorsal ectoderm from our modeling efforts, as we wanted to focus our attention on the areas of the embryo with varying expression levels across mutant constructs.

## Confocal image dataset

For the 59 constructs analyzed, a total of 935 embryo images were taken, with a minimum of 10 images per construct. Late stage 5 (pre-gastrulation) cellularizing embryos were used for analysis, and *eve* expression was used to select the embryos in a narrow age range. Embryos were also selected based on their rotation, so that the *rhomboid* lateral stripe was near the center of the image, with a sufficient number of pixels in the dorsal region of the embryo for background estimation.

## Sequence analysis

Because there are slight differences in the reported PWMs for Dorsal, Twist, and Snail, we considered information from a variety of sources. For Dorsal, PWMs were obtained from two sources: a PWM generated by MEME analysis (with default settings) of footprinted binding sites found in FlyReg (*Bergman et al., 2005*; *Noyes et al., 2008*), herein referred to as DL1, and bacterial one-hybrid experiments (*Zinzen et al., 2006*; *Noyes et al., 2008*; *Ozdemir et al., 2011*), referred to as DL2. The two position probability matrices were then averaged, and the log values calculated from this averaged matrix were used to yield a third hybrid PWM for Dorsal, DL3. For Twist, PWMs were used from two different SELEX experiments (*Zinzen et al., 2006*; *Ozdemir et al., 2011*), herein referred to as TW1 and TW2 respectively. Subsequently, averaging the two PWMs as described above for Dorsal then derived a third hybrid PWM, referred to as TW3. For Snail, three different sources were used: SELEX data from BDTNP (http://bdtnp.lbl.gov), SELEX data from a previously published study (*Zinzen et al., 2006*), and a PWM generated by MEME analysis of footprinted binding sites found in FlyReg (*Bergman et al., 2005*; *Bailey et al., 2009*), herein referred to as SN1, SN2 and SN3 respectively. For analysis of enhancer sequences, we used the MAST program from the MEME software suite to identify putative binding sites (*Zinzen et al., 2006*; *Bailey et al., 2009*). The thresholds used in thermodynamic modeling were evaluated by recovery of known footprinted binding sites, although for some settings not all PWMs were able to find all footprinted sites. P-values for binding sites used in *Figure 2–5* were set at p=0.001 for all factors. PWMs used were: DL1, TW3 and SN3.

## Quantitative data for dorsal, twist, and snail concentrations

Quantitative values for concentrations of Dorsal, Twist and Snail were obtained for early *Drosophila* embryo (stage 5) from a previously published study (*Fakhouri et al., 2010*; *Zinzen et al., 2006*). The published data consisted of 1000 average concentrations for each protein uniformly distributed along the DV axis. Since we were only concerned with the portion of the embryo in the ventral region, we took the region from 0 – 40% of the DV axis and chose a subset of the 1000 data point (17 uniformly distributed data points corresponding to this region; hence a data point every 2.5% of egg height) as our Dorsal, Twist, and Snail concentration gradients. The data used for modeling is given below:

Dorsal: 0.85326 0.77516 0.68914 0.59981 0.51152 0.42792 0.35175 0.28472 0.22757 0.18021 0.14193 0.11165 0.08811 0.07001 0.05618 0.0456 0.03746

Twist: 0.93224 0.88219 0.81279 0.70658 0.54216 0.34085 0.17674 0.08318 0.03873 0.01842 0.00892 0.00433 0.00208 0.00097 0.00044 0.00019 0.00008

Snail: 0.985 0.976 0.967 0.957 0.902 0.441 0.043 0.005 0.001 0 0 0 0 0 0 0 0

## Structure of models

The modeling approach implemented in this study is a thermodynamic-based modeling approach, similar to models used in previous studies (*Zinzen et al., 2006*; *Segal et al., 2008*; *Fakhouri et al 2010*; *He et al., 2010*;). These models are derived using the law of mass action and thermodynamic equilibrium assumptions. They take information regarding the number and arrangement of TF binding sites, as well as TF concentrations, and output predicted levels of gene expression.

Here, we use thermodynamic models that assume RNA polymerase (RNAP) is recruited by bound TFs, and thus model transcriptional output as proportional to the probability of the enhancer being in an 'active state'. Other assumptions used by all models tested in this manuscript include:

- An 'active state' is defined as any state of the enhancer with at least one activator bound and any repressor(s) bound are not actively repressing (quenching) the bound activator(s),

- TF binding affinities are directly proportional to PWM scores obtained using MAST, with one proportionality (scaling) constant per TF,
- interactions (i.e. cooperativity and quenching) only occur between adjacently bound TFs,
- and TFs can not bind simultaneously to overlapping binding sites; competitive binding occurs.

To test different hypotheses about biochemical mechanisms of transcription factor activity on enhancers, several different schemes involving transcription factor cooperativity and short-range repression were implemented in our modeling effort. To create models that considered the diverse cooperativity and repression (referred to as quenching) relationships we propose, all possible pair-wise combinations of the fifteen cooperativity and eight quenching approaches were considered, generating 120 different models.

For short-range repression, we used three continuous functions (Linear-Q2, Logistic-Q3 and Gaussian Decay-Q4) to describe change in repressor activity the percentage of time that the repressor is actively repressing (or quenching) an adjacently bound activator, as a function of the distance, $d$, in base pairs, from the repressor binding site to the activator binding site.

1. Linear $f(d) = a+bd$
2. Logistic Decay $f(d) = 2a/(1+e^{(d/b)})$
3. Gaussian Decay $f(d) = ae^{(-dd/b)}$

When implemented, $a=1$ and $b>0$ is a model parameter for quenching functions. For cooperativity functions, '$a$' and '$b$' are both model parameters. An alternative approach involved 'binning' distances between activators and repressors. We fit quenching parameters (Q) for each of the bins. We also used the non-monotonic 'quenching' function (Q1) derived from our analysis of short-range repression by the Giant protein in synthetic enhancer constructs (*Hansen et al., 2003*; *Fakhouri et al., 2010*; *Suleimenov et al., 2013*).

The binned quenching schemes are described as follows. The distances between binding sites were calculated from the center of the binding sites. Because of minimal center-to-center distances between Snail and Twist or Dorsal, the actual minimal distance possible is 11 bp in the wild-type *rho* enhancer sequence.

Scheme Q5: q1: 1–25 bp, q2: 26–50 bp, q3: 51–75 bp, q4: 76–100 bp
Scheme Q6: q1: 1–35 bp, q2: 36–70 bp, q3: 71–105, q4: 106–140 bp
Scheme Q7: q1: 1–45 bp, q2: 46–90 bp, q3: 91–135, q4: 136–180 bp
Scheme Q8: q1: 1–10 bp, q2: 11–20 bp... q9: 81–90 bp, q10: 91–100 bp

For cooperativity functions, we use the same functions as above (1–3) to describe the multiplicative effect of cooperative binding between two adjacently bound activators, as a function of the distance, $d$, in base pairs, between the activator binding sites. When implemented as cooperativity functions, a>0 and b>0 are both model parameters.

We considered two different ways of estimating cooperativity between transcription factors: heterotypic (between Dorsal and Twist) and homotypic (Dorsal-Dorsal, Twist-Twist, or Snail-Snail). We tested three different continuous functions (Linear-C1, Logistic-C2 and Gaussian Decay-C3), which were parameterized with a single pair of parameters for all homotypic interactions, and separate values for Dorsal-Twist cooperativity. Additional models with 'binned' distances were also considered. For each of the binned schemes, we used a simpler form in which all homotypic interactions are parameterized with the same values, and a more complex form where each type of protein interaction for a given bin size receives distinct parameters. Each of these schemes therefore generates two model forms – binned and protein-binned respectively.

Schemes C4 and C10: c1: 1–25 bp, c2: >25 bp
Schemes C5 and C11: c1: 1–50 bp, c2: >50 bp
Schemes C6 and C12: c1: 1–75 bp, c2: >75 bp
Schemes C7 and C13: c1: 1–50 bp, c2: 51–100 bp, c3: >101 bp
Schemes C8 and C14: c1: 1–60 bp, c2: 61–120 bp, c3: >121 bp
Schemes C9 and C15: c1: 1–70 bp, c2: 71–140 bp, c3: >141 bp
For a summary of parameters in each model, see *Table 1*.

## Parameter estimation

A global parameter estimation strategy, CMA-ES (Covariance Matrix Adaptation - Evolutionary Strategy) was applied to estimate the parameters (*Hansen et al., 2003*; *Segal et al., 2008*;

*Fakhouri et al., 2010*; *Suleimenov et al., 2013*). Root mean square error (RMSE) was used as a measure of performance of different cooperativity and quenching schemes, as described previously (*Matsumoto and Nishimura, 1998*; *Segal et al., 2008*; *Fakhouri et al., 2010*). This RMSE was calculated using the 17 discrete expression points, as described above, and corresponding data points coming from the model predictions. Note that we used these 17 points, taken from 0–40% of the DV axis, and RMSE which gives each point equal weight, because we wanted to focus on the region of the embryo in which varying expression levels were observed and use the same scoring method with no bias across all construct. Due to the stochastic nature of starting points and fixed maximum number of runs for CMA-ES, estimations were run five times, which was empirically found to be sufficient to produce similar, minimal RMSE values for at least three of the runs in at least 47% of cases.

## Model comparison

Since we tested a number of different model formulations, to evaluate the performance of an ensemble of models, we used multiple different approaches. In one approach (results shown in *Figure 7* and *Figure 7—figure supplement 1*), we calculated the average expression profiles of a panel of models on 500 bp fragments from genomic sequences flanking a number of developmental genes. We also investigated ensemble approaches using weighted averages, including an average weighted by the model's performance (one minus the RMSE), and an average weighted by the model's AIC (Akaike Information Criteria), a fitness measure which penalizes for the number of model parameters. The results obtained from these three ensemble approaches were similar. Therefore, only the unweighted average expression profiles are shown in *Figure 7* and and *Figure 7—figure supplement 1*.

## Computations

All image processing was done using ImageJ and MATLAB. Binding site locations were determined using the MAST algorithm in the MEME suite (*Bailey et al., 2009*). All thermodynamic modeling was done using code written in C++ and run on the HPCC (High Performance Computational Cluster) at Michigan State University. Scripts to run MAST, create input files and run multiple versions of the model were written in C++ and Python. The C++ source code for the thermodynamic models used is available at http://www.github.com/arnosti-lab/ThermoModel, the MEME suite is available at http://meme.ebi.edu.au/meme/doc/download.html, and the C-source code for the CMA–ES algorithm is available at http://www.lri.fr/~hansen/cmaes_inmatlab.html#cpp (*Bailey and Gribskov, 1998*; *Hansen et al., 2003*).

## Cross validation

### Systematic cross-validation

Constructs were divided into five sets based on the type of mutation as follows; constructs are numbered as shown in *Supplementary file 1*.

　　Dorsal site knockouts: Constructs 2,3,6,7,8,11,12,15,16
　　Twist site knockouts: Constructs 4,5,17
　　Dorsal and Twist site knockouts: Constructs 9,10,13,14,18,19,20,21
　　Snail site knockouts: Constructs 23–33
　　bHLH site knockouts: Constructs 34–38 (see *Supplementary file 1* for construct details).

　　Parameter estimation was performed using 24 selected models while leaving out data from each of the five sets of constructs. Expression was subsequently predicted for all 38 constructs using parameters obtained, and RMSE over the constructs left out as well as over all 38 constructs was used to analyze the effects of data provided by each set of constructs to the model.

## Random, five fold cross-validation

The 38 constructs to be fitted were separated into 5 randomized partitions of size eight (three partitions) and seven (two partitions). The partitions were computer-generated using the Python random. shuffle method, which is based on the Mersenne Twister algorithm (*Zinzen et al., 2006*; *Matsumoto and Nishimura, 1998*; *Dresch et al., 2010*). This process was repeated five times to give five different partitioning schemes. All 38 constructs were then predicted using parameters from each run, and average RMSE of the constructs left out was considered.

## Sensitivity analysis

Sensitivity analysis was performed for the 24 selected models as previously described (*Zinzen et al., 2006*; *Dresch et al., 2010*). Uninformative parameters, i.e., those with empty bins, were excluded from the analysis. First-order relative sensitivity denotes the sensitivity of model to changes in values of a particular parameter, while second-order sensitivity denotes the sensitivity of model to changes in values of a parameter in combination with other parameters. A parameter with high first-order relative sensitivity is likely to be informative on its own, whereas a parameter with high second-order relative sensitivity implies that the model may have high predictive power, but the parameter values are not informative on their own due to inter-parameter dependencies.

Scoring of predictions from experimentally measured enhancers cloned into pHonda1 that were not included in the model fitting shown in *Figure 6*.

For neurectodermal enhancers, a four-point scheme was applied to score Snail repression as well as neurectodermal activation. Snail repression was measured at nucleus 4. Snail repression was scored as:

1. wild-type, complete repression (expression below 0.1, where 1.0 represents signal in neuroectodermal regions)
2. moderate levels of repression (expression 0.1-0.3)
3. weak repression (expression 0.3-0.5)
4. very weak repression (expression 0.5-1.0)

Neurectodermal activation was scored at the peak in a four-point scheme:

1. Difference between predicted and measured peak expression is less than 0.2
2. Difference between predicted and measured peak expression is between 0.2 and 0.5
3. Difference between predicted and measured peak expression is between 0.5 and 0.7
4. Difference between predicted and measured peak expression is greater than 0.7

For mesodermal enhancers, scoring was done on a five-point scale for activation and four-point scale for Snail activity. The activation score is given below:

1. Mostly mesoderm activation, difference between predicted and measured peak expression is less than 0.2
2. Mostly mesoderm activation, difference between predicted and measured peak expression is between 0.2 and 0.4
3. Mostly mesoderm activation, difference between predicted and measured peak expression is between 0.5 and 0.7
4. Low mesoderm activation, high neurectoderm activation, low dorsal ectoderm activation
5. Low mesoderm activation, high neurectoderm activation, high dorsal ectoderm activation

Snail repression scale is given below:

1. No Snail activity; putative mesoderm activation is equal to >1.5 times peak neurectoderm expression
2. Some Snail activity; mesoderm activation = peak neurectoderm expression
3. High snail activity; mesoderm activation< neurectoderm activation
4. Highest snail activity; low expression in mesoderm (<0.1 intensity value)

## Acknowledgements

We acknowledge the Michigan State University Institute for Cyber Enabled Research and John Johnston in the High Performance Computing Center for assistance with computational analysis, Dr. Melinda Frame (MSU Center for Advanced Microscopy) for assistance with confocal microscopy, Nicholas Panchy for help with sequence analysis, Anne Sonnenschein, Rewatee Gokhale, Max Winkler and Ramona Beckman for assistance with cloning of constructs, Dr. Robert Zinzen and Dr. Dmitri Papatsenko for help with DVEx dataset, Arnosti laboratory members for helpful discussions, Dr. Yuehua Cui for advice on statistical analysis, Martin Scherr for assistance with Mathematica coding, and Dr. Ahmet Ay (Colgate University) for comments on the manuscript. This study was supported by NIH GM056976 to DNA and a fellowship from the MSU Quantitative Biology Program to JMD.

## Additional information

### Funding

| Funder | Grant reference number | Author |
|---|---|---|
| National Institutes of Health | GM056976 | David N Arnosti |
| MSU Quantitative Biology Program | Graduate student fellowship | Jacqueline M Dresch |

The funders had no role in study design, data collection and interpretation, or the decision to submit the work for publication.

### Author contributions

RS, JMD, Conception and design, Acquisition of data, Analysis and interpretation of data, Drafting or revising the article; IP, Acquisition of data, Analysis and interpretation of data, Drafting or revising the article; BRT, Acquisition of data, Analysis and interpretation of data; DNA, Drafting or revising the article

### Author ORCIDs

David N Arnosti, http://orcid.org/0000-0003-0983-6982

## Additional files

### Supplementary files

• Supplementary file 1. Description and sequence information for constructs List of all constructs with numbers, names, and sequences. Restriction sites are capitals, with red capital letters indicating mutated binding site sequences.

• Supplementary file 2. Expression data for constructs Quantification of signal intensity along the DV axis of the embryo with standard error for each construct. Sheet 2 has the 17 data points used for fitting on each construct.

• Supplementary file 3. Parameters for all model runs List of parameters for all five runs of each model.

### Major datasets

The following previously published datasets were used:

| Author(s) | Year | Dataset title | Dataset URL | Database, license, and accessibility information |
|---|---|---|---|---|
|  |  | BDTNP SELEX | http://bdtnp.lbl.gov/Fly-Net/selex.jsp | Publicly available at Berkeley Drosophila Transcription Network Project |
| Bergman CM, Carlson JW, Celniker SE | 2005 | Flyreg | http://bergmanlab.ls.manchester.ac.uk/flyreg/ | Publicly available at Drosophila DNase I Footprint Database |

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
