## [Decision Letter]

Thank you for submitting your work entitled "Quantitative perturbation-based analysis of gene expression predicts enhancer activity in early *Drosophila* embryo" for peer review at *eLife*. Your submission has been favorably evaluated by Naama Barkai (Senior editor), a Reviewing editor, and three reviewers.

The reviewers have discussed the reviews with one another and the Reviewing editor has drafted this decision to help you prepare a revised submission.

Summary:

Sayal et al. report extensive experimental and computational studies of rho-expression in the dorsal-ventral patterning gene regulatory network of *Drosophila*. The aim is to understand how rho-expression depends on binding sites of transcription factors (Twist, Snail, Dorsal) in the rho enhancer. To this end, constructs with individual transcription factor binding sites removed are analyzed and the resulting patterns of rho expression fitted to different models of transcription factor binding, cooperation, and activity.

Essential revisions:

1) The reviewers and the BRE are concerned about the multi-model analysis. They all agree that this analysis is unclear and non-standard. The reviewers were also concerned about the procedure over-fitting to the data. Select comments attached below. Clarification and possibly rethinking the mode of analysis is needed for the revision. One of the main claims of the paper is the ability to estimate biophysical parameters, this claim has to be convincingly demonstrated.

A few comments from the reviewers on this issue:

"The concept of ensemble of models is not new and is the subject both of Bayesian statistics (which is essentially about learning how much we can trust each possible model) and non-Bayesian formulations (e.g., mixture-of-experts models). I suspect that employing one of these methodologies would actually lead to a useful predictive rule which might apply on the non-rho enhancers. Right now the discussion that points to the models that support the observations is weak."

"Based on their cross-validation results, the individual models are overfit. They claim that fitting a suite of models addresses this because they can identify trends amongst models. However, the set of models is highly related; they vary only in the particular pair-wise interactions that are allowed and the way that the distance-dependence is encoded. I believe it's more appropriate to address overfitting using sensitivity analysis, which they did. Sensitivity analysis should reveal which subset of parameters are well constrained by the data and which are not. But their conclusion is that cooperativity and quenching parameters show extensive mutual dependence, leaving it unclear precisely which parameters they can interpret across models."

"The authors find extensive "parameter compensation", meaning that the data does not constrain the model parameters, implying generally a good model fit but poor predictive performance (unless the instances where prediction is sought are similar to examples already seen during model training). Given the large number of parameters, one would expect a good fit (and also a good cross-validation on examples similar to training data) even if the overlap between models and properties of the enhancer, or models among themselves is poor."

"Possibly the question how impressive the predictions are can be settled on the basis of the present data. The overlap between predicted and observed expression levels can be plotted on a scatter plot against distance of the enhancer configuration to the most similar configuration in the training set. Results would differ somewhat depending on what distance measure is used, but I expect one can obtain insight into how serious the overfitting is."

2) With regards to the experimental data, rigorous measurement of expression levels by typical in situ is not trivial. The analysis here measured differences in expression level by keeping the acquisition settings on the microscope constant and normalizing maximum expression to a WT stain imaged on the same day. However, there can also be significant error from batch to batch variation in probes and stains, which are not accounted for here. This issue should be addressed by examining batch to batch variability and demonstrating that it's not an issue for their system, or by simulating data with more uncertainty in the levels of expression and see if this qualitatively effects their results.

[Editors' note: further revisions were requested prior to acceptance, as described below.]

Thank you for resubmitting your work entitled "Quantitative perturbation-based analysis of gene expression predicts enhancer activity in early *Drosophila* embryo" for further consideration at *eLife*. Your revised article has been favorably evaluated by Naama Barkai (Senior editor), a Reviewing editor, and three reviewers. The manuscript has been improved but there are some remaining issues that need to be addressed before acceptance. This includes in particular the minor comments of reviewer #1. In addition, in response to reviewer #2 comment, please be more explicit about the main result that was learnt from your analysis (beside the general modeling success)

Reviewer #1:

The authors have diligently responded to the previous comments and the manuscript is more clear as a result. Overall, I find the main conclusions – that activators cooperate in a distance independent way while repressors cooperate less and have distance dependent function relative to activators – to be quite interesting. The modeling strategy is somewhat unusual, and while it's presented fairly clearly in the text, I don't find the figures always give great insight. However, for most aspects, I think readers can assess what they did and draw their own conclusions about the validity of the modeling approach and whether it provided useful insights. I do not see any technical errors that would undermine their main conclusions, and I think they are appropriately circumspect. I can therefore support publication.

Reviewer #1 (Minor Comments):

I still have minor concerns about how their imaging data is being analyzed and incorporated into the model. These are mostly issues of clarification and how to make best use of the dataset that they have in hand.

First, in response to my previous concern about the variation in their imaging data, the authors quantified variability in their control embryos, which were stained in different batches and imaged on different days, and used to calibrate the acquisition settings of their microscope, thus serving as the effective normalization for peak expression for their test constructs. They analyze the variation in this full set, reporting that their mean peak expression was 138+/- 36.8, and that background was 52.4 +/- 23.8. They are thus seeing about 2X expression above background for their WT controls across all stains by this metric. This is a useful number and it's good to include (I would leave out the qualitative description of it's meaning that follows, but that's a personal preference for numbers over verbal description).

However, if increases in background and signal are correlated (for example if you have embryos that are brighter or dimmer overall, which is often the case with fluorescent stains), this might be actually be over-estimating their error. Instead, or in addition, they could include the standard error for the WT images used to calibrate the microscope for each construct in Figure 1, instead of plotting only the average WT trace across all of these data without any error. It's possible that they did not collect sufficient data on the WT constructs during each imaging session, in which case this wouldn't be helpful. An alternative would be to plot the standard error for the average WT trace after normalization.

I'm looking for some clarification on the imaging primarily because many of their constructs decrease or move expression moderately and it's helpful to know whether any of those constructs fall within experimental error. In the methods, it states that they imaged each test construct on a single day, so it's not possible to do the bulk analysis of error described above for each construct. The standard error for each construct shown in Figure 1 is therefore representative of the variation in their dataset. Including the error for WT in Figure 1 would largely address this issue.

I appreciate that they introduced noise into their predictions to try to assess how their error in expression measurement might impact their conclusions. This reveals the sensitivity of the top models, and their likely over-fitting, which confirms the utility of not focusing on a single model, but it is not necessary to include this analysis in this already large paper.

Second, to clarify how the expression measurements feed into their models, the authors could include a more explicit description of the inputs, outputs and scoring scheme for their models in the section "Thermodynamic modeling of *rho* enhancer perturbation dataset". For example, did they use continuous expression data as their input, or select expression levels at specific points? The latter seems to be the case based on Figure 2 where black dots are specific measurements. How/why were these points selected? The output also appears to be at discrete points based on this Figure, where red lines are predictions and I am assuming that the line is from linking together these discrete points. Finally, the RMSE score they use equally weights errors in regions of high expression and low expression. Some alternative scores have been used in other work to emphasize other features of expression patterns (for example the Sinha group has used the weighted Pattern Generating Potential). It is not necessary to score their models differently, but they should discuss the trade-offs of their particular scoring scheme. And how is the RMSE score calculated? Is it done on the continuous data (red lines in Figure 2 vs. continuous expression measurements), or on the point by point data as above? I could not find this information in the main text or the methods. I suggest that it is included in the main text as a verbal description of their procedures, and any additional information be put in the methods. It also seems possible and potentially useful to include their experimental error in Figure 2, to allow comparison of biological variability against their model performance.

Reviewer #2:

The revised version of Sayal et al. has improved in some ways. Overall I am supportive of this type of work, which involves careful experimental measurements followed by an honest attempt to fit the data to a physically inspired model. The strengths of the paper are the quantitative data, and the attempt to validate the model predictions both by testing the activity of enhancers from other species and by using the model to predict the locations of other enhancers, some of which are already known.

The main weakness of the paper remains the decision to center the analysis around a collection of 120 related models rather than attempting to find the best model, or a small number of competing models. This gives the analysis a very unsystematic feel; different models are highlighted and put forward in different parts of the paper. The result is that the claims of the paper are vague (that experiments and models can help reveal regulatory principles) rather than something concrete about the grammar of this system. There is no clear statement of what the authors have learned besides the idea that modeling is a good thing.

In my opinion the issue of over fitting has been dealt with adequately by the experiments in which the activity of orthologous enhancers are predicted and measured and by the demonstration that there is little correlation between the accuracy of new predictions and the similarity of the tested construct to constructs in the training data.

Reviewer #3:

The additional checks requested in the previous round of review alleviate my concern concerning overfitting. My assessment is that this work goes beyond a resource of detailed experimental data, and the statistical analysis unveils aspects of the underlying biology. Detailed analysis of the data can be done in the future and is beyond the scope of this paper. For this reason I recommend publication in *eLife*.

---

## [Author Response]

*Essential revisions:*

*1) The reviewers and the BRE are concerned about the multi-model analysis. They all agree that this analysis is unclear and non-standard. The reviewers were also concerned about the procedure over-fitting to the data. Select comments attached below. Clarification and possibly rethinking the mode of analysis is needed for the revision. One of the main claims of the paper is the ability to estimate biophysical parameters, this claim has to be convincingly demonstrated. Few comments from the reviewers on this issue: "The concept of ensemble of models is not new and is the subject both of Bayesian statistics (which is essentially about learning how much we can trust each possible model) and non-Bayesian formulations (e.g., mixture-of-experts models). I suspect that employing one of these methodologies would actually lead to a useful predictive rule which might apply on the non-rho enhancers. Right now the discussion that points to the models that support the observations is weak."*

A major question concerned our presentation and analysis of the set of models utilized in this study. In the original manuscript, we described the individual fit and performance of a number of models that test cooperativity, activity and quenching (local repression). As the reviewers point out, a common way of evaluating such ensembles is using average or weighted averages to generate a hybrid model that may summarize the overall performance. Following the reviewers’ suggestion, we tested three different schemes for an ensemble approach, a simple average, an RMSE-score weighted average, and an AIC penalized approach. The simple average ensemble model, now shown in revised Figure 7 and Figure 7—figure supplement 1, generated predictions that were similar to the RMSE-weighted and AIC penalized ensemble models (Figure 8).

Author response image 1.Predictions of twenty four *rho*-fit thermodynamic models for the *brinker* genomic region.Model structure indicated by nomenclature indicating cooperativity/quenching schemes used, and overall ranking in fitting for *rho* training set. Red indicates neuroectodermal-like pattern; blue mesodermal-like pattern prediction. *brk* transcription unit indicated by gray box. Green peaks indicate ChIP-seq measured binding of regulators Twist, Snail and Dorsal. Gray bars indicate experimentally validated enhancers. Boxes below indicate (top to bottom) simple averages of ensemble of models with standard deviation, RMSE weighted averages, and AIC weighted averages. For these predictions, the simple average closely resembled the other two methods..**DOI:**
http://dx.doi.org/10.7554/eLife.08445.020

We now note in the revised Results that this approach provides a convenient metric for transcriptional potential predictions, yet the graphic display of the individual models also serves a role, depicting those regions where there is a high degree of uniformity about the prediction (e.g. primary and shadow *brk* enhancers) and where the models diverge sharply (*eve* proximal region). We also now cite and discuss a paper that was recently published from Saurabh Sinha’s group, in which a different ensemble approach is used to identify regulatory logic of the *ind* enhancer (Hassan Samee et al., Cell Systems). Their approach differs from ours in that a single thermodynamic model structure is utilized, and the different models correspond to unique parameter sets that were found to provide good fits under cis and trans perturbation.

*"Based on their cross-validation results, the individual models are overfit. They claim that fitting a suite of models addresses this because they can identify trends amongst models. However, the set of models is highly related; they vary only in the particular pair-wise interactions that are allowed and the way that the distance-dependence is encoded. "*

An additional question concerned model structure and overfitting. We do see that performance for individual models drops as we leave out portions of the training dataset, but this is mostly apparent when we partition the training set in a non-random manner. As shown in Figure 2—figure supplement 1, models performed well with five-fold random cross validation, but leaving out all of the data from specific classes of perturbations (e.g. removal of repressor sites) had a significant impact on RMSE. We include the “nonrandom” partitioning to indicate how important different types of perturbations are; some studies rely on saturation mutagenesis to achieve the same effect, but testing thousands of variants is impractical in this system, so a targeted survey was called for (White et al., 2013). We have revised the Results section to emphasize this point. As the reviewer noted, the structures of the models are related, which may lead to similarities in identified parameters. We agree that this similarity of models may indeed produce such effects; the reason that we did not try vastly different models is that certain aspects of rho binding proteins are experimentally known, based on extensive empirical research. For instance, we know that Snail is a short-range repressor, thus we used models in which it would be possible to recover such activity, depending on estimated parameters. (However, we could also recover parameters that would be consistent with long-range action, depending on recovered parameters for longer distance intervals). Similarly, strong evidence indicates that Twist and Dorsal are activators, and can show cooperative action, thus models in which the proteins had inverted regulatory activities (as explored by Jaeger and Reinitz on less well-characterized settings) were not included. Our models are not too narrowly structured, merely filling in some parameters for highly detailed biochemical mechanisms, however; complex processes of transcriptional control are subsumed in the single “activity” parameter, which is an appropriate simplifying role for models. Given these limitations, the diversity of the models that was generated helped identify general trends that point to possible important biological properties. For example, we noted a substantial increase in performance for models in which we separately assess activator and repressor cooperativity, indicating that these proteins may have different cooperative potentials.

*"The authors find extensive "parameter compensation", meaning that the data does not constrain the model parameters, implying generally a good model fit but poor predictive performance (unless the instances where prediction is sought are similar to examples already seen during model training). Given the large number of parameters, one would expect a good fit (and also a good cross-validation on examples similar to training data) even if the overlap between models and properties of the enhancer, or models among themselves is poor."*

We did indeed point out in the manuscript that the different model formulations of activation and repression were interdependent, likely a result of parameter compensation, and provide direct evidence for parameter compensation in Figure 4—figure supplement 1 i-ii, where Twist activity parameters are higher on one run, correlated with lower homotypic Twist-Twist cooperativity values. The presence of some degree of compensation is neither surprising, nor does it invalidate the approach; depending on the extent of such compensatory interactions, models will be more or less overfit, which will come out as the models are applied to novel situations, tested later in the manuscript.

*“I believe it's more appropriate to address overfitting using sensitivity analysis, which they did. Sensitivity analysis should reveal which subset of parameters are well constrained by the data and which are not. But their conclusion is that cooperativity and quenching parameters show extensive mutual dependence, leaving it unclear precisely which parameters they can interpret across models."*

We did indeed conduct sensitivity analysis to evaluate the extent to which parameters of different models were informative as well as to determine which were constrained. A low sensitivity overall would indicate that the parameter was not at all informative, while high first-order sensitivities would indicate that a parameter is uniquely informative, and not compensating with other values. We find that most parameters showed high overall sensitivity, indicating their importance in driving model fitting, and high second-order values, indicating that there are interdependencies between these factors (only the Snail activity parameter showed high first-order sensitivities in some models). This latter finding of higher second-order sensitivities is not at all surprising, and it was a factor that we recognized embarking on this study, because the mechanism of transcription involves processes that all contribute to the final output, and research on native biological systems has demonstrated how transcriptional output can be set by complementary contributions of cooperativity or factor potency, for instance. We were interested in determining how useful such models would be despite some degree of parameter interdependence, which is borne out by success in predicting novel enhancers. We do think that the clustering of many parameter values across many models is biologically informative. For instance, the tendency for long-range activator cooperativity vs. low repressor cooperativity is likely to be informative, but this must be tested. We revised the Discussion to note that structural similarities of models may also drive such effects, as the reviewer pointed out.

*"Possibly the question how impressive the predictions are can be settled on the basis of the present data. The overlap between predicted and observed expression levels can be plotted on a scatter plot against distance of the enhancer configuration to the most similar configuration in the training set. Results would differ somewhat depending on what distance measure is used, but I expect one can obtain insight into how serious the overfitting is."*

Following the reviewer’s suggestion, we tested this idea by plotting the predictions for different novel enhancer sequences against the similarity of the closest element in the training set. (Figure 9) The similarity of regulatory sequences was calculated by a standard word-count algorithm from the Vingron laboratory used previously to examine the similarity of metazoan regulatory regions. The plot does not show a strong positive correlation between this simple measure of similarity and modeling success. This result indicates that the simple measure of enhancer relatedness by global feature similarity is insufficient to predict activity. Indeed, the features that dictate enhancer output are very specific and context-dependent. Experimental work has shown individual cases in which subtle changes in a few residues are sufficient to strongly affect enhancer output, while at the same time highly divergent DNA sequences can mediate essentially identical tissue-specific expression of different genes (Crocker and Erives, 2008).

Author response image 2.Average RMSE (suite of 24 models) vs enhancer distance as calculated using N2 method.There is no strong trend toward increased performance (lower average RMSE) with more closely related enhancers. N2 method run using code provided in Göke J et al. (Bioinformatics, 2012).**DOI:**
http://dx.doi.org/10.7554/eLife.08445.021

2) With regards to the experimental data, rigorous measurement of expression levels by typical in situ is not trivial. The analysis here measured differences in expression level by keeping the acquisition settings on the microscope constant and normalizing maximum expression to a WT stain imaged on the same day. However, there can also be significant error from batch to batch variation in probes and stains, which are not accounted for here. This issue should be addressed by examining batch to batch variability and demonstrating that it's not an issue for their system[…]”

As noted above, we used internal controls to normalize data acquired in each imaging session. In response to this comment, we re-examined the primary data acquired for the modeling, looking at control gene expression used for day-to-day normalization. Differences in probe bleaching, laser intensity, gain settings of the CCD etc. will all impact overall signal intensity. Following the reviewer’s suggestion, we reanalyzed 348 control images captured over 53 imaging sessions. We find that for these images of embryos with enhancers containing the wild-type ensemble of activators, the range of average peak intensities, prior to any background subtraction or normalization, was 56.8 – 255 units (only 3 were at 255, saturation value). The mean was 138. +/- 36.8 (S.D.). Thus, for the large majority of captured signals, the day-to-day differences in intensities were not very great, and normalization procedures were not changing values by large factors. The background signals from non-expressing portions of the embryos were 52.4 +/- 23.8 (S.D.), thus considerably below the signal; and in all cases the strong signals measured on any day were well above background measured on any day. We have now included information about this range and variation of signal intensities in the Materials and methods.

“[…]or by simulating data with more uncertainty in the levels of expression and see if this qualitatively effects their results.”

As suggested here, we also carried out parameter estimation with 5% or 10% noise added to the measured gene expression (Figure 10). As shown below, the top ranked models were generally more strongly affected, while midranked models were more modestly impacted. Such behavior provides an indication of the degree of overfitting for these models. (The lowest ranked model improved with the noisy data sets, consistent with our observation that for this model, the original parameter estimation runs apparently did not achieve a global minimum). It is interesting that the predictions on genomic loci such as *brk* (not used in model fitting) are consistent with the “top” models revealing their overfitting – they did not out performing mid-level models in many cases.

Author response image 3.RMSE for each of 24 models run with data simulating 5% noise and 10% noise.Ten datasets were generated to randomly introduce 5% or 10% noise, respectively, into the input data. The model was then fit on each of these noisy datasets as before, and the average RMSE obtained from the 10 datasets at 5% noise and 10% noise is plotted here, as well as the RMSE obtained with initial model fitting..**DOI:**
http://dx.doi.org/10.7554/eLife.08445.022

[Editors' note: further revisions were requested prior to acceptance, as described below.]

*Reviewer #1 (Minor Comments):*

*I still have minor concerns about how their imaging data is being analyzed and incorporated into the model. These are mostly issues of clarification and how to make best use of the dataset that they have in hand. First, in response to my previous concern about the variation in their imaging data, the authors quantified variability in their control embryos, which were stained in different batches and imaged on different days, and used to calibrate the acquisition settings of their microscope, thus serving as the effective normalization for peak expression for their test constructs. They analyze the variation in this full set, reporting that their mean peak expression was 138+/- 36.8, and that background was 52.4 +/- 23.8. They are thus seeing about 2X expression above background for their WT controls across all stains by this metric. This is a useful number and it's good to include (I would leave out the qualitative description of it's meaning that follows, but that's a personal preference for numbers over verbal description).*

We agree that this number is important and have added this (2X) to the paragraph in which we address the variation in the wild-type signal in the Methods section.

*However, if increases in background and signal are correlated (for example if you have embryos that are brighter or dimmer overall, which is often the case with fluorescent stains), this might be actually be over-estimating their error. Instead, or in addition, they could include the standard error for the WT images used to calibrate the microscope for each construct in Figure 1, instead of plotting only the average WT trace across all of these data without any error. It's possible that they did not collect sufficient data on the WT constructs during each imaging session, in which case this wouldn't be helpful. An alternative would be to plot the standard error for the average WT trace after normalization.*

We appreciate the reviewer bringing this to our attention. We agree that showing the standard error for the WT construct (which was very small) as well as the other constructs in Figure 1 improves the figure and illustrates the variation in our dataset. We have, therefore, modified Figure 1. Now, instead of showing a single embryo’s *rho* expression, it shows the average WT signal after normalization and includes the standard error.

*I'm looking for some clarification on the imaging primarily because many of their constructs decrease or move expression moderately and it's helpful to know whether any of those constructs fall within experimental error. In the methods, it states that they imaged each test construct on a single day, so it's not possible to do the bulk analysis of error described above for each construct. The standard error for each construct shown in Figure 1 is therefore representative of the variation in their dataset. Including the error for WT in Figure 1 would largely address this issue. I appreciate that they introduced noise into their predictions to try to assess how their error in expression measurement might impact their conclusions. This reveals the sensitivity of the top models, and their likely over-fitting, which confirms the utility of not focusing on a single model, but it is not necessary to include this analysis in this already large paper.*

As indicated by the reviewer, this additional analysis is confirmatory in nature, but adds little to the overall story, and we agree with the suggestion to leave this analysis out of the manuscript.

*Second, to clarify how the expression measurements feed into their models, the authors could include a more explicit description of the inputs, outputs and scoring scheme for their models in the section "Thermodynamic modeling of rho enhancer perturbation dataset". For example, did they use continuous expression data as their input, or select expression levels at specific points? The latter seems to be the case based on Figure 2 where black dots are specific measurements. How/why were these points selected? The output also appears to be at discrete points based on this Figure, where red lines are predictions and I am assuming that the line is from linking together these discrete points. Finally, the RMSE score they use equally weights errors in regions of high expression and low expression. Some alternative scores have been used in other work to emphasize other features of expression patterns (for example the Sinha group has used the weighted Pattern Generating Potential). It is not necessary to score their models differently, but they should discuss the trade-offs of their particular scoring scheme. And how is the RMSE score calculated? Is it done on the continuous data (red lines in Figure 2 vs. continuous expression measurements), or on the point by point data as above? I could not find this information in the main text or the methods. I suggest that it is included in the main text as a verbal description of their procedures, and any additional information be put in the methods.*

The data used for model fitting was indeed from 17 discrete data points. To clarify how the expression data was used in our model fitting, we have added a detailed description of how we discretized the continuous data to the Methods section.

We have also added a verbal description of what data was used to calculate the RMSE, as well as our reasoning behind using this method.

*It also seems possible and potentially useful to include their experimental error in Figure 2, to allow comparison of biological variability against their model performance.*

The experimental error is shown for seven different constructs in Figure 1. This error is representative of the error found in all constructs tested. Due to this fact, we have decided not to include the error in Figure 2, as we feel it will add little to the overall message, while reducing the readability of the figure.

*Reviewer #2:*

*The revised version of Sayal* et al. *has improved in some ways. Overall I am supportive of this type of work, which involves careful experimental measurements followed by an honest attempt to fit the data to a physically inspired model. The strengths of the paper are the quantitative data, and the attempt to validate the model predictions both by testing the activity of enhancers from other species and by using the model to predict the locations of other enhancers, some of which are already known.*

*The main weakness of the paper remains the decision to center the analysis around a collection of 120 related models rather than attempting to find the best model, or a small number of competing models. This gives the analysis a very unsystematic feel; different models are highlighted and put forward in different parts of the paper.*

As indicated in our previous revision, there are indeed different aspects to using an ensemble model, as opposed to exploring the significance of the 120 models presented in Figure 2. We explicitly tested the importance of exact spacing relationships for the activators by generating models that permitted fine-scale differentiation of protein binding sites, however, most models with separate repressor and activator cooperativity parameters tended to high, and distance-independent positive cooperativity for activators, as illustrated in Figure 3. We subsequently focus on a single subset of 24 models for sensitivity and cross-validation, showing that broad properties apply across this set of models (and doubtless the entire 120 related models), such as demonstration that certain types of perturbations (e.g. hitting both activator and repressor sites) were critical for optimal fitting, that the overall depth of perturbation was sufficient for obtaining quantitative information about the “transcriptional grammar” of this enhancer, and that on most models, parameter was observed. Thus, in the end we make our major conclusions about two groups of models; the entire set, and the representative subset of 24 (as well as the demonstration that an “average” ensemble captures some, but not all useful information about novel genes that are scanned in Figure 5.

*The result is that the claims of the paper are vague (that experiments and models can help reveal regulatory principles) rather than something concrete about the grammar of this system. There is no clear statement of what the authors have learned besides the idea that modeling is a good thing.*

We agree that this aspect of the paper must be clearer. Following the reviewer’s suggestion, we have summarized the key findings of our paper, along with comments as to how modeling approach provides concrete, testable hypotheses, in the Discussion section.